



# Insights into secondary organic aerosol formation from the day- and nighttime oxidation of PAHs and furans in an oxidation flow reactor

Abd El Rahman El Mais[1,2], Barbara D'Anna[2], Luka Drinovec[3], Andrew T. Lambe[4], Zhe Peng[5], Jean-Eudes Petit[6], Olivier Favez[1], Selim Aït-Aïssa[1], Alexandre Albinet[1]

[1]Ineris, Parc Technologique Alata, Verneuil-en-Halatte, 60550, France
[2]Aix Marseille Univ, CNRS, LCE, Marseille, France
[3]Center for Atmospheric Research, University of Nova Gorica, Nova Gorica, Slovenia
[4]Aerodyne Research, Inc. (ARI), Billerica, Massachusetts, USA
[5]CIRES and Department of Chemistry, University of Colorado, Boulder, Colorado 80309, USA
[6]Laboratoire des Sciences du Climat et de l'Environnement (LSCE), 91190 Gif sur Yvette, France

*Correspondence to:* Alexandre Albinet (alexandre.albinet@ineris.fr)

**Graphical abstract.**

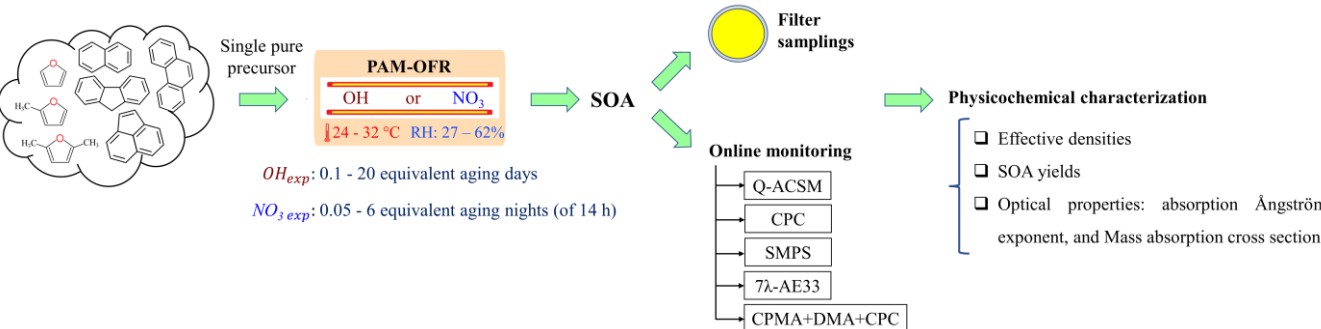

**Abstract.** Secondary organic aerosols (SOA) formed by oxidation of typical precursors largely emitted by biomass burning, such as PAHs and furans, are still poorly characterized in terms of formation yields, physical and light absorption properties, particularly those generated at night following reaction with nitrate radicals ($NO_3$). In the present study, we evaluated and compared the formation yields, effective density ($\rho_{eff}$), absorption Ångström exponent ($\alpha$), and mass absorption coefficient (MAC) of laboratory-generated SOA from three furan compounds (furan, 2-methylfuran, and 2,5-dimethylfuran) and four PAHs (naphthalene, acenaphthylene, fluorene, and phenanthrene). SOA were generated in an oxidation flow reactor from the reaction between hydroxyl radicals (OH; 0.1 - 20 equivalent aging days) or $NO_3$ radicals (0.05 - 6 equivalent aging nights of 14 h) with single furan or PAH. The $\rho_{eff}$, formation yields, $\alpha$, and MAC of the generated SOA varied depending on the precursor and oxidant considered. The $\rho_{eff}$ of SOA formed with OH and $NO_3$ tended to increase with particle size before reaching a "plateau". This was particularly evident for the nighttime chemistry experiments with $NO_3$ radicals (1.2 to 1.6 on average for particles $\geq$ 100 nm). Such results highlighted potential differences in the chemical composition of the SOA, as well as probably in their morphology, according to the particle size. Three times lower SOA formation yields were obtained with $NO_3$ compared



to OH. The yields of PAH SOA (18 to 76 %) were 5 to 6 times higher than those obtained for furans (3-12 %). While furan SOA showed low or negligible light absorption properties, PAH SOA was found to have a significant impact in the UV-Visible region, implying a significant contribution to atmospheric brown carbon (BrC). No increase in the MAC values was observed from OH to $NO_3$ oxidation processes, probably due to a low formation of nitrogen-containing chromophores through

homogeneous gas phase oxidation processes with $NO_3$ only (without $NO_x$). Overall, the results obtained in this work demonstrated that PAHs are significant precursors of SOA emitted by biomass burning, through both, day- and nighttime processes, and have a substantial impact on the aerosol light absorption properties and so probably on climate.

**Keywords:** Secondary organic aerosols (SOA), Formation yields, Density, Brown carbon, Biomass burning.

## 1 Introduction

Airborne particles (particulate matter, PM) have strong impacts on air quality, and climate (Monks et al., 2009; Fuzzi et al., 2015; IPCC, 2022). They have also been associated with adverse health effects, including respiratory and cardiovascular diseases, mortality, and morbidity (Pope et al., 2002; Brook et al., 2010; Thurston et al., 2016; Rajagopalan et al., 2018; Pope et al., 2020). Biomass burning is one of the major sources of fine PM ($PM_{2.5}$) in ambient air, particularly in the winter season due to residential wood combustion (RWC) used for heating purposes (Vicente and Alves, 2018; Crippa et al., 2013; Weber

et al., 2019; Zhang et al., 2019, 2020; Chen et al., 2017; Srivastava et al., 2018b; Denier van der Gon et al., 2015; Chen et al., 2022a; Viana et al., 2016). This source also emits large amounts of volatile and semi-volatile organic compounds (VOCs and SVOCs) (Akherati et al., 2020; Bruns et al., 2016; Ahern et al., 2019; Hartikainen et al., 2018; Růžičková et al., 2022; Hatch et al., 2015, 2017, 2018; Baudic et al., 2016) that can undergo (photo-) chemical oxidation processes involving atmospheric oxidants, such as ozone ($O_3$), hydroxyl (OH) or nitrate ($NO_3$) radicals, resulting in the formation of Secondary Organic

Aerosols (SOA) (Kroll and Seinfeld, 2008; Hallquist et al., 2009; Jimenez et al., 2009; Ziemann and Atkinson, 2012; Carlton et al., 2009; Heald and Kroll, 2020). OA constitute a significant fraction of fine PM (Bressi et al., 2021; Zhang et al., 2007, 2011), and SOA account for a large proportion of OA (up to 90%, depending on the location) (Zhang et al., 2007, 2011; Srivastava et al., 2018a; Kroll and Seinfeld, 2008). Identifying the major SOA precursors and investigating the physicochemical properties, formation yields, and chemical composition of SOA are crucial to implement efficient air quality

policies.

Among the different VOCs and SVOCs emitted from RWC, some have been identified as major SOA precursors, such as benzene, toluene, phenols, furans, and polycyclic aromatic hydrocarbons (PAHs) (Yee et al., 2013; Bruns et al., 2016; Tiitta et al., 2016; Ahern et al., 2019; Akherati et al., 2020; Růžičková et al., 2022). The reactivity of PAHs and furans through homogeneous (in gaseous phase) or heterogenous (gas/particle) processes with different oxidants is well documented in the

literature (Keyte et al., 2013; Jiang et al., 2020; Li et al., 2018; Ringuet et al., 2012; Aschmann et al., 2011, 2014; Al Ali et al.,



2022; Bierbach et al., 1995; Newland et al., 2022; Kind et al., 1996; Zheng et al., 2006). However, the level of scientific understanding in terms of SOA formation from such species is still limited. Different studies available in the literature have been focused on the formation yields, chemical composition, and physicochemical properties of SOA formed from PAHs and furans (Lee and Lane, 2009; Chan et al., 2009; Lee and Lane, 2010; Lee et al., 2012; Shakya and Griffin, 2010; Kleindienst et
al., 2012; Zhou and Wenger, 2013a, b; Chen et al., 2016; Riva et al., 2017; Kautzman et al., 2010; Gómez Alvarez et al., 2009; Strollo and Ziemann, 2013; Jiang et al., 2019b; Joo et al., 2019a; Tajuelo et al., 2021; Srivastava et al., 2022; Joo et al., 2019b; Jiang et al., 2019b; Chen et al., 2022b; Jiang et al., 2019a). However, most of these studies have been conducted with OH and/or $O_3$ radicals, and only a small number of them have investigated the SOA formation with nitrate radicals ($NO_3$). While OH and $O_3$ are the key atmospheric oxidants during the day, $NO_3$ is known to be the major one during the night (Brown and
Stutz, 2012). The study of $NO_3$ radical chemistry is crucial for describing winter pollution, when RWC is more relevant, and when the night lasts longer than the day. Several authors have reported its significance in the formation of SOA from RWC emissions (Kodros et al., 2020; Jorga et al., 2021; Kodros et al., 2022; Tiitta et al., 2016). The current underestimation of OA concentrations by air quality models by a factor of 3 to 5 in winter might be partly due to the neglected nighttime chemistry (Fountoukis et al., 2016; Mircea et al., 2019; Tsimpidi et al., 2014). Recent simulation results including this nighttime
chemistry have shown that more than 70% of the OA from biomass combustion are significantly influenced by the oxidation processes that take place in the absence of light, i.e., involving nitrate radical (Kodros et al., 2020).

Finally, atmospheric brown carbon (BrC) aerosols, which primarily absorb light in the shorter visible to the ultraviolet (UV) wavelengths, have been recognized to play a critical role in the Earth's radiative balance (Hems et al., 2021; Laskin et al., 2015; Moise et al., 2015; Andreae and Gelencsér, 2006). As for non-absorbing OA species, BrC can also impact black carbon
(BC) light absorption due to the so-called lensing effect under internal mixing conditions (Zhang et al., 2018; Saleh et al., 2015). On the other hand, the aging of BrC-containing OA may result in decreasing their absorptivity (Sumlin et al., 2017). Quantifying the BrC contribution to light absorption is therefore essential for an accurate interpretation of the aerosol optical depth (AOD), the atmospheric column's light extinction due to both scattering and absorption, and regional climate. Numerous studies have quantitatively characterized the parameters governing the optical absorption properties (e.g. absorption angström
exponent (AAE, α), Mass Absorption Coefficient (MAC), and Refractive index) of SOA formed from various biogenic and anthropogenic precursors (Lambe et al., 2013; Liu et al., 2016; Xie et al., 2017a; Dingle et al., 2019; Siemens et al., 2022; Laskin et al., 2015; Jiang et al., 2019a; Metcalf et al., 2013; Klodt et al., 2023; He et al., 2022; Chen et al., 2022b; Hems et al., 2021; Moise et al., 2015), but the information regarding SOA formed from PAHs (other than naphthalene) and furans is still quite scarce (e.g., (Cheng et al., 2020)). Besides, as particle density is linked to optical properties due to its dominant role in
the effects on refractive index, its accurate determination is crucial for the evaluation of SOA radiative forcing (Liu and Daum, 2008). Overall, density is a key physical property of particles because it influences transport properties and so, the fate of particles in both the atmosphere and the human respiratory system (Seinfeld and Pandis, 1998; Finlayson-Pitts and Pitts Jr, 2000).





The main objectives of this study were to evaluate and compare the formation yields, physical (granulometry and effective
densities ($\rho_{eff}$)), and light absorption properties (α and MAC) of the SOA formed from typical precursors emitted by biomass
burning, namely PAHs and furans following OH and NO$_3$-initiated oxidating aging processes.

## 2  Experimental methods

### 2.1  Generation of radicals and SOA using OFR

Most of the previous studies focusing on the SOA formation from PAHs and furans have been conducted in environmental
(smog) chambers. Oxidation Flow Reactors (OFR) are an alternative tool to simulate atmospheric aging (Peng and Jimenez,
2020). Several works have shown the reliability, comparability, and complementary of the results provided using such systems
in the study of SOA formation processes (Bruns et al., 2015; Peng and Jimenez, 2020). Their application in the study of SOA
formation from key precursors emitted by biomass burning is of prime interest (Hodshire et al., 2019; Srivastava et al., 2022).
In our study, the stable (up to 17 hours) generation of SOA (Figure S1 of the Supplementary Material, SM) has been carried
out at ambient temperature (T = 24 - 32 °C) and environmentally relevant relative humidity (RH = 25 - 62 %) in a Potential
Aerosol Mass – Oxidation Flow Reactor (PAM-OFR, Aerodyne Research) (Kang et al., 2007; Lambe et al., 2011), which is a
13.3 L aluminum horizontal cylindrical chamber (46 cm long x 22 cm ID) operated in a continuous flow mode (Figure 1).
Three furans (furan (99% purity), 2-methylfuran (2-MF, 99%) and 2,5-dimethylfuran (2,5-DMF, 99%)), and four PAHs
(naphthalene (Naph, 99%), acenaphthylene (Acy, 75%, the remaining 25% are acenaphthene (Ace)), phenanthrene (Phe, 99%),
and fluorene (Flu, 98%)) have been selected as SOA precursors and studied here. All were purchased from Sigma Aldrich
(France) and injected into the PAM-OFR in their pure state, in the absence of NO$_x$, and without seed particles. The injection
method was determined by the state of the different substances under ambient temperature conditions. Liquid furans were
injected, at a constant flow, through a 0.0152 cm i.d. Teflon tubing using a microliter syringe pump (TriContinent C24000,
100 μL, syringe plunger velocity of 3 to 17 μsteps s$^{-1}$ (= 5.63 to 31.88 μL min$^{-1}$) depending on the compound studied and
concentration generated) and subsequently nebulized/vaporized using a clean air flow of 2 L min$^{-1}$ adjusted by a Mass Flow
Controller (MFC, MKS G series) before being introduced into the PAM-OFR. PAHs were vaporized by circulating heated
clean air through a metallic tube packed with the corresponding solid compounds (50-3500 mL min$^{-1}$, depending on the PAH
studied and concentration generated, regulated at 30 ºC using a CROCO-CIL liquid chromatography column oven). A filtered
air supply (TSI 3074B) combined with an air zero generator (Claind AZ air purifier 2010) were used to deliver clean-dry air
and maintain the total carrier gas flow inside the PAM-OFR at 9.5 L min$^{-1}$.



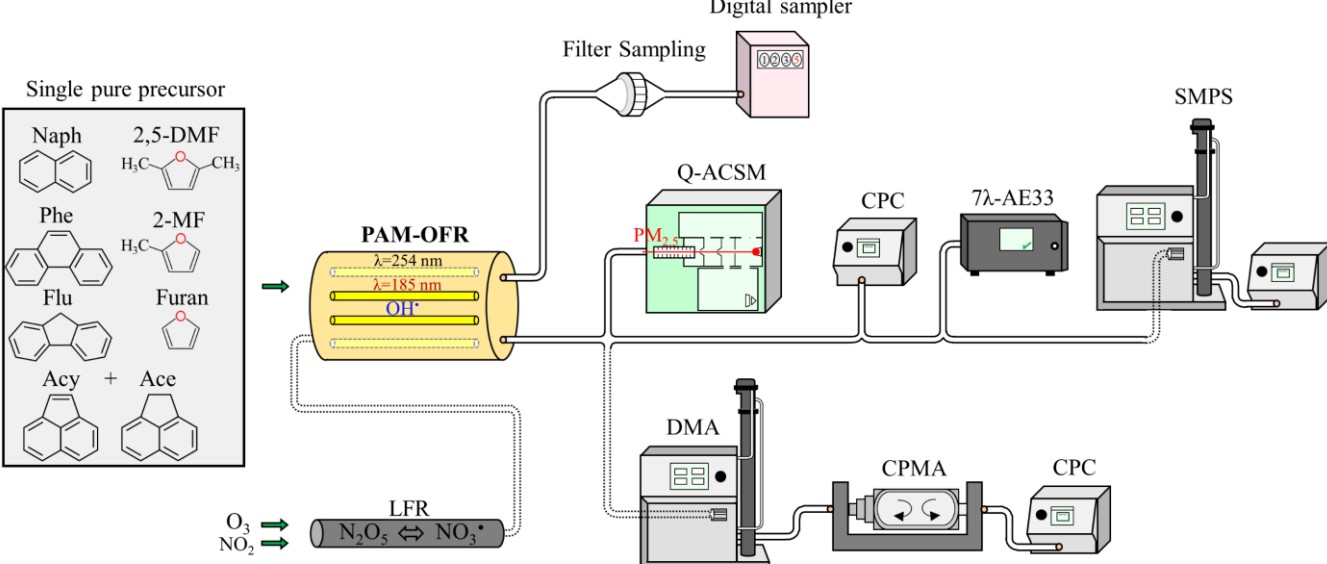

**Figure 1. Simplified schematic diagram of the experimental set-up used to generate and monitor the resulting SOA.**

To simulate the daytime chemistry, OH radicals were produced by photolysis of $O_2$ by four UV lamps emitting at $\lambda = 185$ and
254 nm (OFR185 method) (Li et al., 2015). Briefly, the photolysis of $O_2$ at 185 nm produces $O_3$ which is then photolyzed at
254 nm to produce $O(^1D)$. $O(^1D)$ then reacted with water vapor (introduced using a Nafion membrane humidifier, Perma Pure
LLC, FC100-80-6-MKK) to generate OH radicals. The lamps' voltage was set, depending on the precursor studied, between
1.8 and 3 V, and the corresponding irradiance (Ir), ranging between 6 and 126 $\mu W$ cm$^{-2}$ (Table 1) was continuously measured
by the PAM-OFR photodiode (Tocon_C6, sglux). A UV photometric ozone analyzer (Model 202, 2B Technologies) has been
used to monitor the $O_3$ concentrations in the PAM-OFR.

For nighttime chemistry, $NO_3$ radicals were generated from the thermal decomposition of $N_2O_5$ at room temperature in a dark
OFR (OFR-iN$_2$O$_5$ method;(Lambe et al., 2020). Briefly, $NO_3$ and $N_2O_5$ were generated in the gas phase following the reaction
of separate reagent flows containing approximately 100 ppm $NO_2$ (1% in $N_2$, Air Liquide) and 200 ppm $O_3$ in a 152.4 cm
long × 2.22 cm ID perfluoroalkoxy (PFA) laminar flow reactor (LFR) coupled to the PAM-OFR. To achieve these reagent
concentrations, the $NO_2$ flow (1 % in $N_2$) was set at 20 cm$^3$ min$^{-1}$, and 2 L min$^{-1}$ of pure $O_2$ (99.9995%, Air Liquide) was
passed through an ozone chamber housing a mercury fluorescent lamp. The $O_3$ mixing ratio inside the LFR was measured
using an additional ozone analyzer (Model 202, 2B Technologies). In such conditions, $NO_2$ concentrations in the PAM-OFR
(NO$_X$ Analyzer 42i-HL, Thermo Scientific) were below the detection limit.





**Table 1. Experimental conditions used to study the SOA formation from selected PAHs and furans under daytime conditions (OH radical, OFR185).**

| SOA precursor | SOA precursor concentration $[VOC]_0$ (mg m$^{-3}$) | Lamps voltage (V) | Ir ($\mu$W cm$^{-2}$) | RH (%) | T ($^\circ$C) | $[O_3]_{PAM}$ (ppm) | Equivalent aging days | |
| | | | | | | | Experimental evaluation | KinSim calculations |
|---|---|---|---|---|---|---|---|---|
| 2-Methylfuran | 8.99 ± 0.20 | 3.0 | 71 | 38 | 30 | 15 | 6 | 9 |
| 2,5-Dimethylfuran | 17.68 ± 0.02 | 2.8 | 63 | 38 | 29 | 12 | 3 | 6 |
| Furan | 18.41 ± 0.05 | 2.5 | 34 | 34 | 29 | 7 | nd[1] | 0.1 |
| Naphthalene | 2.60 ± 0.17 | 1.8 | 6 | 41 | 26 | 3 | 3 | 2 |
| Acenaphthylene | 0.33 ± 0.02 | 1.8 | 14 | 39 | 30 | 4 | 8 | 12 |
| Fluorene | 0.43 ± 0.01 | 3.0 | 70 | 39 | 29 | 11 | 17 | 20 |
| Phenanthrene | 0.19 ± 0.02 | 3.0 | 58 | 27 | 29 | 10 | 11 | 15 |

[1]not determined.

**Table 2. Experimental conditions used to study the SOA formation from selected PAHs and furans under nighttime conditions (NO$_3$ radical, OFR-iN$_2$O$_5$).**

| SOA precursor | SOA precursor concentration $[VOC]_0$ (mg m$^{-3}$) | $[O_3]_{0, LFR}$ (ppm) | $[NO_2]_{0,LFR}:[O_3]_{0,LFR}$ | RH (%) | T ($^\circ$C) | $[O_3]_{PAM}$ (ppm) | Equivalent aging nights of 14 h |
|---|---|---|---|---|---|---|---|
| 2-Methylfuran | 41.80 ± 0.10 | 194 | 0.52 | 49 | 26 | 11 | 0.1 |
| 2,5-Dimethylfuran | 50.23 ± 0.11 | 179 | 0.56 | 53 | 25 | 6 | 0.1 |
| Furan | 18.44 ± 0.00 | 200 | 0.5 | 62 | 26 | 10 | * |
| Naphthalene | 15.10 ± 0.11 | 194 | 0.51 | 43 | 25 | 12 | 5.7 |
| Acenaphthylene | 3.05 ± 0.10 | 183 | 0.55 | 49 | 25 | 13 | 1.7 |
| Fluorene | 1.07 ± 0.10 | 198 | 0.5 | 45 | 26 | 8 | ** |
| Phenanthrene | 0.26 ± 0.01 | 200 | 0.5 | 43 | 25 | 10 | 5.6 |

*Unstable furan injection inducing an unstable SOA generation.

**No SOA generation observed.

## 2.2 Estimation of the injected SOA precursor concentrations

Furans and PAH concentrations introduced into the PAM-OFR were established in the range of 8 to 50 mg m$^{-3}$ and 0.19 to 15.1 mg m$^{-3}$, respectively (Tables 1 and 2) to produce and collect sufficient quantities of PM (filter samplings at about 6.5 L min$^{-1}$) for further *in vitro* biological responses assessment.

Furan concentrations were calculated from the syringe pump injection flow rate, temperature, analyte molecular weight, density, and dilution ratio into the OFR carrier gas. PAH concentrations were evaluated during spare experiments under the same conditions as for SOA generation, but without oxidants (PAM-OFR lights off, and no O$_3$ and NO$_2$ injected into the LFR, only O$_2$). Particulate and gaseous phases were collected at the PAM-OFR exit on quartz fiber filters (Pallflex Tissuquartz, Ø = 47 mm) and polyurethane foams (PUFs; Tisch Environmental, 1.5 x 3 inches), respectively. Prior to sampling, filters were pre-heated at 500 °C for 12 h, while PUFs were pre-cleaned using pressurized solvent extraction (ASE 350, Thermo; one



hexane cycle followed by two acetone cycles: 80 °C, 100 bars, 5 min heat time, 15 min static time) (Zielinska, 2008). Sampling durations were 15 and 30 min, and experiments were performed in duplicate for each duration. One or two field blanks (one for each sampling duration), for each PAH and oxidant, were also collected with no PAH injection into the PAM-OFR. After collection, samples were wrapped in aluminum foils, placed in zip bags, and then stored at -20 °C until analysis. The filter and

its associated PUF (28 samples and 12 blanks in total) were extracted together using pressurized liquid extraction (Dionex, ASE 200, 80 °C, 100 bars, 5 min heat time, 15 min static time, 2 cycles) with acetonitrile as the solvent (VWR, HPLC grade). Prior to extraction, a known amount of 6-methylchrysene was added to the samples and used as a surrogate standard to check the extraction efficiency (ranging from 88 to 108%). The extracts were then directly quantified (no reduction step to avoid any loss of PAH by evaporation, only a 10 times dilution was applied) by UPLC-Fluorescence-UV (Ultimate 3000, Thermo

Scientific) using a C18 UPLC column (Zorbax Eclipse PAH, 2.1 mm × 150 mm × 1.8 µm, Agilent, 1 µL injected).

### 2.3 Online SOA characterization

The SOA generated have been monitored and characterized using a set of different online instrumentations (Figure 1). Organic aerosol mass and chemical composition of the non-refractory aerosol fraction were measured using a Quadrupole Aerosol Chemical Speciation Monitor (Q-ACSM; Aerodyne Research Inc.), equipped with a $PM_{2.5}$ aerodynamic lens, using a 1 min

time resolution. The aerosols were dried before analysis using a Nafion dryer system. Calibration of the detector response factor was performed by using ammonium nitrate and sulfate solutions (Ng et al., 2011; Crenn et al., 2015; Freney et al., 2019). A Relative ion efficiency (RIE) of 1.4 was applied to organic matter. The instrument was equipped with a capture vaporizer so that a fixed collection efficiency (CE) of 1 could be used for the whole ACSM dataset (Xu et al., 2017). Connected in series with the Q-ACSM, a Condensation Particle Counter (CPC; Grimm version 5.403), and a Scanning Mobility Particle Sizer

(SMPS, composed of a Differential Mobility Analyzer (DMA) 3081, an electrostatic classifier 3080, and a CPC 3775) were used to monitor in parallel the total particle number concentration (in the range of 4.5 nm to 3 µm, 1 s time resolution) and the particle size distribution (from 14.6 to 661.2 nm, 5 min time resolution). SOA effective density ($\rho_{eff}$) was evaluated by combining a DMA, a Centrifugal Particle Mass Analyzer (CPMA, Cambustion) (Olfert and Collings, 2005; Olfert et al., 2006; Johnson et al., 2013) and a CPC, all connected in series with the Q-ACSM (instead the SMPS). The mass of the generated

SOA was determined for a given particle size (mobility diameter) over a range of 30 to 200 nm allowing the evaluation of the SOA $\rho_{eff}$ according to the particle size. Such an approach has already been used to determine the effective density of particles of different types of primary emissions (vehicular, flame soot) and SOA (Peng et al., 2021; Malloy et al., 2009), but only few data exist for SOA formed from anthropogenic and biogenic precursors. Finally, a 7-wavelengths (370, 470, 520, 590, 660, 880, and 950 nm) aethalometer (AE33, Magee scientific, 1 min time resolution) was also connected at the exit of the PAM-

OFR. Such online filter tape-based optical measurement method is commonly used for the measurement of black carbon concentrations (BC, at 880 nm) in ambient air as well as for the determination of BrC aerosol fraction (Drinovec et al., 2015; Zhang et al., 2020; Drinovec et al., 2017). Here it was used to tentatively assess the SOA light absorption properties and to





compare the results obtained with the most conventional techniques in the literature such as photoacoustic spectrometry or on aerosol sample extract by UV-Vis spectroscopy (Moise et al., 2015).

## 2.4 Calculations

### 2.4.1 OH and NO₃ exposures

The OH exposure (OH$_{exp}$) was determined experimentally by continuously measuring the decay of SO$_2$ injected into the PAM-OFR (200 ppb as initial concentration) during spare experiments with the same conditions as for SOA generation, using a SO$_2$ analyzer (AF 21 M Environment S.A) (Lambe et al., 2015) and applying Eq. (1)

$$OH_{exp} = -\frac{1}{k_{SO_2}^{OH}} \times ln\left(\frac{[SO_2]}{[SO_2]_0}\right) \qquad (1)$$

where $k_{SO_2}^{OH}$ is the rate constant of the reaction between OH and SO$_2$, equal to 9.4 x 10$^{-13}$ cm$^3$ molecule$^{-1}$ s$^{-1}$ (Davis et al., 1979). [SO$_2$]$_0$ was the initial SO$_2$ concentration injected into the PAM-OFR, and [SO$_2$] was the final concentration after oxidation. The OH exposure ranged from 3.74 x 10$^{11}$ to 2.19 x 10$^{12}$ molecules cm$^{-3}$ s (Table S1), corresponding to 3 - 17 equivalent aging days, respectively (Table 1), assuming a daily average OH radical concentrations of $1.5 \times 10^6$ molecules cm$^{-3}$ (Finlayson-Pitts and Pitts Jr, 2000; Mao et al., 2009).

In addition to the experimental evaluation, OH$_{exp}$ was evaluated using the KinSim chemical kinetic solver (Peng and Jimenez, 2019). Inputs to the OFR KinSim model included Pressure (P), T, RH, [VOC]$_0$ (Table 1), total residence time ($\tau$ = 84 s), photon fluxes at 185 and 254 nm ($I_{185}$ and $I_{254}$, respectively) (Table S2), bimolecular rate constants of each precursor with OH or O$_3$, and photoabsorption cross sections ($\sigma_i$) of each precursor at 185 and 254 nm (Table S3).

The obtained OH$_{exp}$ ranged from 1.05 x 10$^{10}$ to 2.57 x 10$^{12}$ molecule cm$^{-3}$ s (Table S1), corresponding to about 0.1 - 20 equivalent aging days, and were comparable, even though larger, with the experimental values (Table 1). These differences seemed to be due to a significant rise in OH concentrations after the VOC consumption, which proportionally influenced the modeled OH$_{exp}$ values (Figure S2).

NO$_{3exp}$ was estimated theoretically using a KinSim mechanism adapted from previous work (Lambe et al., 2020). Model parameters included P, T, RH, [VOC]$_0$, [O$_3$]$_{0, LFR}$ (Table 2), $\tau$ of 109 s, [NO$_2$] of 100 ppm, and the bimolecular rate constants of each precursor with NO$_3$ or O$_3$ (Table S3). NO$_{3exp}$ values ranged from $1.68 \times 10^{12}$ to $2.10 \times 10^{14}$ molecule cm$^{-3}$ s (Table S1), corresponding to about 0.05 to 5.7 equivalent nights of 14 h (Table 2)  at an average nighttime NO$_3$ mixing ratio of 30 ppt (Asaf et al., 2010). Again, most of the NO$_3$ exposure was due to the increase of NO$_3$ concentrations after VOC consumption (especially in the cases of Acy, 2-MF, and 2,5-DMF, Figure S2).





### 2.4.2 Effective density

The effective density ($\rho_{eff}$) of SOA particles was calculated using Eq. (2) as the ratio of the measured particle mass (corresponding to the mode in the scan data ($m_p$)) to the calculated particle volume, assuming a spherical particle (shape factor = 1) with a diameter equals to the electrical equivalent mobility diameter ($d_m$) selected by the DMA (McMurry et al., 2002; DeCarlo et al., 2004):

$$\rho_{eff} = \frac{6}{\pi}\frac{m_P}{(d_m)^3} \qquad (2)$$

### 2.4.3 SOA yields

SOA yields were calculated using Eq. (3):

$$SOA\ yield = \frac{mass\ of\ SOA\ produced}{mass\ of\ VOC\ reacted} \qquad (3)$$

where the mass of VOC reacted with the corresponding precursor was obtained from the KinSim model (see section 3.1), and the mass of SOA was determined from Q-ACSM measurements as the SOA generation was stable (Figure S1). SOA was considered equivalent to OM for OH radical experiments, and equal to OM + NO$_3$ for NO$_3$ radical experiments. Measurement of particle transmission through the PAM-OFR, in the range of 30 – 200 nm, using ammonium sulfate aerosols showed particle

loss below 5 % for all PM sizes, therefore no particle wall-loss correction was applied.

### 2.4.4 Optical absorption coefficient ($b_{abs}$), absorption Ångström exponent (α), and Mass absorption coefficient (MAC)

The AE33 has been designed for automatic correction of the so-called filter loading effect (Drinovec et al., 2015, 2017). Briefly, the sampled ambient air is divided, and the sample is deposited onto two filter spots at different flow rates, leading to uneven loadings on the respective filter spots. A compensation parameter k(λ) is retrieved from the different loading effect

magnitudes influencing these two spots, and k(λ) is further used to determine the light attenuation due to carbonaceous aerosols However, initial results obtained, mainly for the PAH SOA, showed a poor compensation efficiency and large jumps during the spot changes. Manual compensation of AE33 data was therefore applied using a fixed filter loading compensation parameter k(λ) (Table S4) to get usable data (Figure S3).

Light absorption coefficients ($b_{abs}$) were calculated from the manually compensated AE33 values according to Eq. (4).

$$b_{abs}(\lambda) = BC\ (\lambda) \times \sigma_{air}(\lambda) \times \frac{1.41}{3.75} \qquad (4)$$

where $\sigma_{air}$ is the air mass absorption efficiency (Table S5). Comparison between AE33 and photothermal interferometer PTAAM-2λ showed that a higher value of multiple scattering parameter C should be used (Drinovec et al., 2022) than the one provided by the manufacturer. A new C value of 3.75 was selected for the measured particle sizes.

Absorption Ångström exponents (α) were determined as the absolute value of the slope of ln($b_{abs}$) as a function of ln (λ) from 370 to 590 nm. The fitting parameters are summarized in Table S6.





Mass absorption coefficients (MAC) were finally calculated as the ratio of $b_{abs}$ to the SOA mass concentrations evaluated from Q-ACSM measurements using Eq. (5).

$$MAC\ (\lambda) = \frac{b_{abs}\ (\lambda)}{SOA\ mass\ concentration} \tag{5}$$

## 3 Results and discussion

### 3.1 Understanding of oxidants (and photolysis) competition in the PAM-OFR for both oxidation methods

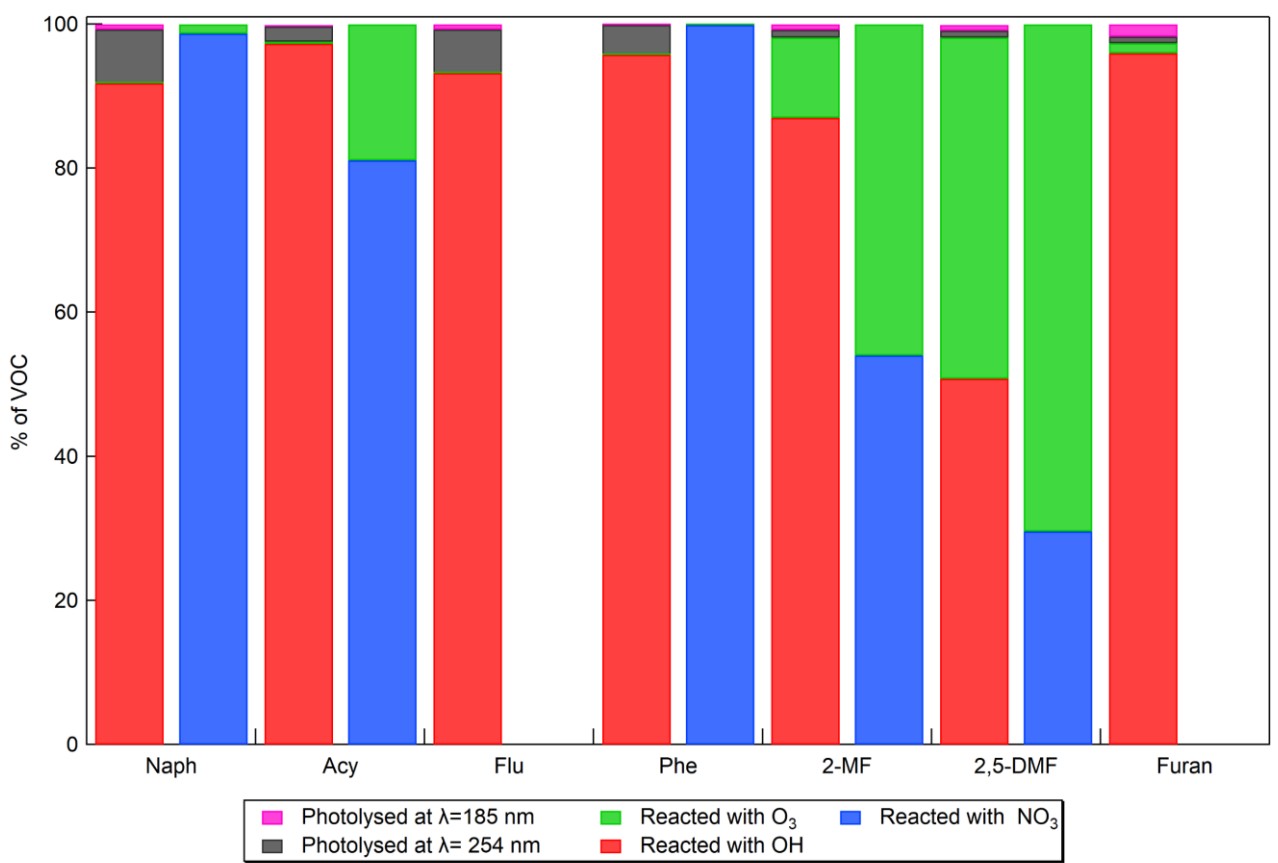

**Figure 2. Relative importance of PAH and furan loss pathways inside the PAM-OFR during both OFR185 (daytime) and OFR-iN₂O₅ (nighttime) modes. No results are shown for Furan and Flu with NO₃ radicals due to an unstable SOA generation or no SOA formed.**

Figure 2 shows the competition between the oxidants, as well as direct photolysis, on the reactivity of the PAHs and furans studied inside the OFR.

In the case of OFR185 (OH reactivity), the predominant reaction of PAHs was observed with OH radicals (> 92%). The remaining photolyzed at 254 nm. Both ozonolysis and photolysis at 185 nm had negligible impact on the reactivity of PAHs.



For furans, photolysis at both wavelengths was negligible (1%), and the competition between OH and $O_3$ differed depending on the compound considered. In the case of furan, ozonolysis was negligible, and the majority (96%) reacted with OH. This contrasted with 2-MF and 2,5-DMF, where $O_3$ competed with OH to consume 11% of 2-MF, and nearly half of 2,5-DMF.

In the OFR-i$N_2O_5$ mode ($NO_3$ reactivity), only $O_3$ and $NO_3$ were in competition. For Naph and Phe, $O_3$ had no impact. However, ozonolysis is significant for Acy (+ 25% of Ace) (20%) or even predominant for 2-MF (46%) and 2,5-DMF (70%). These results showed that in our experimental conditions, the reaction with OH or $NO_3$ were largely the predominant ones, except for 2-MF and 2,5-DMF. These results were used to calculate the exact concentrations of the SOA precursor that reacted to get the most accurate evaluation of the SOA yields.

## 3.2  SOA granulometry and effective density

Figure 3 shows the average number-weighted mobility size distributions obtained for the generated SOA under both day- and nighttime oxidation processes.

Overall, the observed particle size distributions were all monomodal and showed a shift towards larger particle sizes according to the precursor concentrations injected into the PAM-OFR. For instance, for daytime chemistry, the injected concentrations

of Phe, Acy, and Flu were about 0.19 to 0.43 mg m$^{-3}$, and the resulting SOA particle distributions were centered around 32 nm; while the particle size distributions for Naph, 2-MF, 2,5-DMF, and Furan were centered around 55 and up to 100 nm due to the larger precursor concentrations injected (2.6 to 18 mg m$^{-3}$) (Table 1). Similarly, for $NO_3$ radical experiments, as the precursor concentrations of Naph, Acy, and 2-MF were 4.6 to 10 times larger than during OH radical experiments (Table 2), a strong shift in the size distribution towards larger particles, centered around 200 nm, was observed. Interestingly, for similar

precursor concentrations, the SOA formed from Phe by reaction with $NO_3$ also shifted to a large particle size of about 85 nm. However, for 2,5-DMF, the particle size distribution shifted towards smaller particles ($D_p$ ~ 35 nm) even though the concentration injected was about 3 times higher than during OH exposure studies.



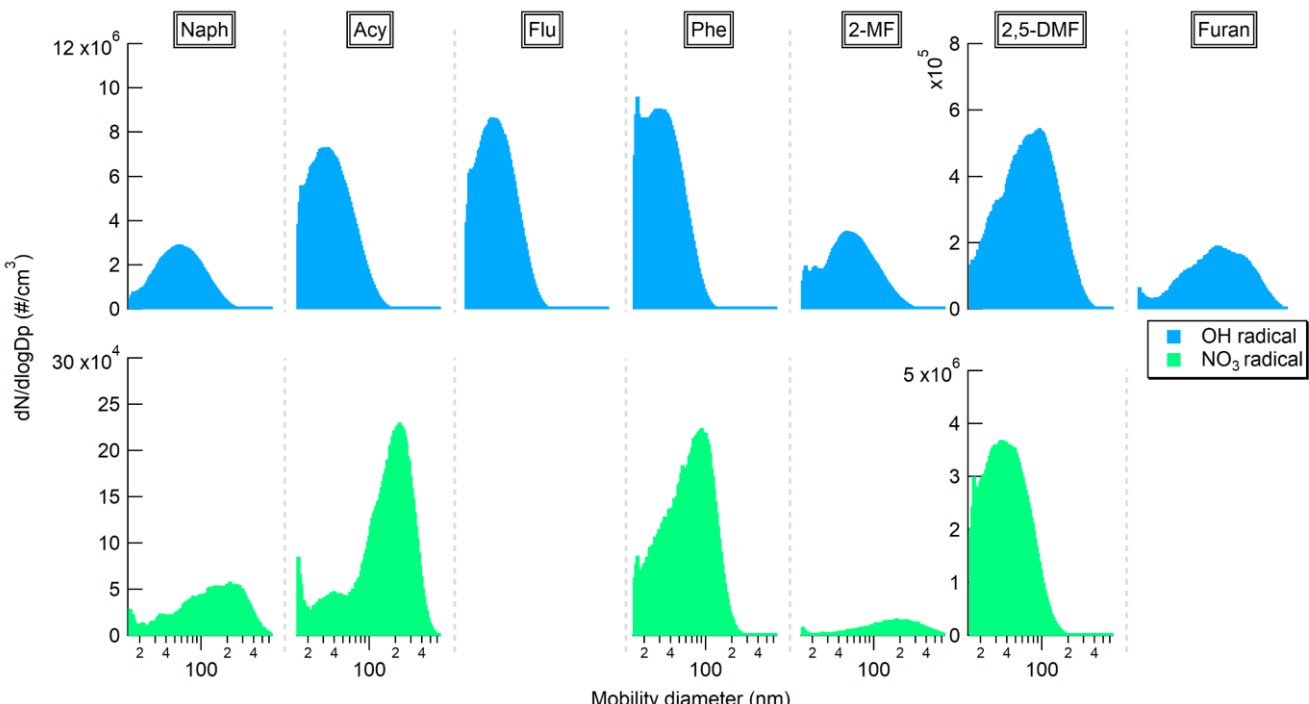

**Figure 3. Comparison of the particle size distributions, in number, of the PAHs and furans SOA formed under day and nighttime conditions (OH and NO₃ radicals respectively). No results are shown for Furan and Flu with NO₃ radicals due to unstable SOA generation or no SOA formed.**

The effective densities of the generated SOA with OH and NO₃ radicals as a function of the aerosol mobility diameter are presented in Figure 4.

Generally, the SOA $\rho_{eff}$ tended to increase with the particle size in the range of 30 to 100 or 150 nm for both reactivities. A "plateau" was then observed especially in the case of the SOA formed by reaction with NO₃ radical. These results highlighted potential differences in the chemical composition of the SOA, as well as probably in terms of their morphology, with the particle size which should be further investigated.

The average $\rho_{eff}$ for each precursor are presented in Table 3. Values for particle diameter ≥ 100 nm were considered according to the "plateau" observed and assuming that this range of particles contributes significantly to the particle mass. Relatively large variations can be observed depending on the precursor and oxidant being studied, ranging from 1.23 to 1.61 g cm⁻³. Between OH and NO₃ oxidation, the effective densities of Naph and Acy SOA increased from 1.33 to 1.39 g cm⁻³ and from 1.23 to 1.47 g cm⁻³, respectively. In contrast, the effective densities of Phe and 2-MF SOA decreased from 1.52 to 1.29 and to 1.38 g cm⁻³, respectively, while they remained comparable for 2,5-DMF. The measured Naph SOA $\rho_{eff}$ were consistent with previously reported values (1.35-1.55 g cm⁻³) from smog chamber experiments at low and high NOₓ conditions (Chan et al., 2009; Chen et al., 2016, 2018; He et al., 2022). In addition, densities of 1.24 and 1.40 g cm⁻³ have been respectively reported





for 3-methylfuran SOA (Joo et al., 2019a) and furan SOA (Chen et al., 2022b), both formed by reaction with NO$_3$. Overall the values determined here were in agreement with previously reported results for SOA formed from different anthropogenic precursors (1.06-1.65 g cm$^{-3}$) (Nakao et al., 2013; Kostenidou et al., 2007; Hallquist et al., 2009). In the absence of density measurements, an SOA density of 1.4 g cm$^{-3}$ is usually assumed for smog chamber experiments (Shakya and Griffin, 2010;

Riva et al., 2017; Jiang et al., 2019b; Hallquist et al., 2009). Our results confirm this default value as a fairly good approximation, but also that it may need to be refined according to the given investigated compounds, for instance, for determining the particle transport properties, and converting PM size distributions into mass concentrations (McMurry et al., 2002). Therefore, and as reported before, our study shows the importance of figuring out the density for individual experiments to evaluate the formation yields (Chen et al., 2016), as well as to assess the optical properties of the studied SOA.


**Table 3. Average effective densities of the PAHs and furans SOA formed under day and nighttime conditions (OH and NO$_3$ radicals respectively). Results obtained from the mean of the densities corresponding to SOA with $d_m$ > 100 nm.**

| Precursors | Aerosol effective densities (g cm$^{-3}$) | |
| --- | --- | --- |
| | OH radical | NO$_3$ radical |
| Naphthalene | 1.33 ± 0.11 | 1.39 ± 0.06 |
| Acenaphthylene | 1.23 ± 0.05 | 1.47 ± 0.13 |
| Fluorene | 1.60 ± 0.10 | No SOA formation |
| Phenanthrene | 1.52 ± 0.02 | 1.29 ± 0.11 |
| 2-Methylfuran | 1.52 ± 0.06 | 1.38 ± 0.06 |
| 2,5-Dimethylfuran | 1.35 ± 0.08 | 1.35 ± 0.08 |
| Furan | 1.61 ± 0.05 | Not determined |



**Figure 4.** Comparison of the effective density of the PAHs and furans SOA formed under day and nighttime conditions (OH and NO₃ radicals respectively) as a function of the aerosol mobility diameter. The error bars correspond to the standard deviations (2 σ) on the number of measurements replicates (n= 2 to 4). No results are shown for Furan and Flu with NO₃ radicals due to unstable SOA generation or no SOA formed.





### 3.3 SOA yields

Figure 5 presents the SOA yields for the studied PAHs and furans under day- and nighttime oxidations as a function of the radical exposure used for each precursor.

The SOA yields from the oxidation of PAHs with OH radicals were all comparable, especially for Naph, Acy, and Flu, ranging from 62 to 76 %. The differences observed, including the highest SOA yield for Phe, could be related to the differences in OH exposures ranging from 2.78 x $10^{11}$ to 2.57 x $10^{12}$ molecules cm$^{-3}$ s (Table S1). The same has been observed for furans with

comparable SOA yields (9-12 %) even if the range of tested OH exposures was largely variable (from 1.05 x $10^{10}$ to 1.13 x $10^{12}$ molecule cm$^{-3}$ s). For NO$_3$-initiated SOA formation, approximately 3 times lower SOA yields were observed relative to OH-initiated SOA formation for most precursors. Only Acy showed an average SOA yield of 44 % with NO$_3$ comparable to that observed with OH radical chemistry (68 %) even though the NO$_3$ exposure tested was the lowest of the PAH group. Table 4 compares the calculated SOA yield with the literature. Most of the previous works reported smog chamber experiments with

OH radicals, under low or high NO$_X$ regimes and dry conditions (RH < 10 %) (Chan et al., 2009; Shakya and Griffin, 2010; Kleindienst et al., 2012; Chen et al., 2016; Riva et al., 2017; Chen et al., 2018; Gómez Alvarez et al., 2009; Jiang et al., 2019b; Tajuelo et al., 2021). For the NO$_3$ radical with furan, only one previous study has reported an SOA yield of 7 % (Jiang et al., 2019a). (Joo et al., 2019a) reported SOA yields of 1.6 to 2.4% for 3-methylfuran in smog chamber dry experiments. As the experimental conditions are quite different, the reported SOA yields are also highly variable, yet our results lie on the upper

range of previously reported works. Our results are 2 to 3 times higher than those of the single study performed using an OFR (Wang et al., 2018) for which no information on OH exposure is available. Other factors may influence the SOA yields such as NO$_x$, seed particles, [VOC]$_0$, temperature, and/or RH used which can enhance the SOA formation by aqueous physicochemical processes (Srivastava et al., 2022; Lambe et al., 2015). Overall, SOA yields from furans oxidation were about 5-6 times lower than those from PAHs, implying that the contribution of furans to SOA formation from biomass burning

emissions seems limited, whereas the PAHs contribution is probably significant during both, day- and nighttime periods.





**Figure 5. Comparison of the PAHs and furans SOA yields obtained from the day- and nighttime oxidation processes (with OH and NO₃ radicals respectively). Results evaluated using Q-ACSM measurements and expressed as a function of OH or NO₃ exposures obtained from KinSim simulations. No results are shown for Furan and Flu with NO₃ radicals due to unstable SOA generation or no SOA formed.**



**Table 4. Comparison of the SOA yields obtained for the different PAHs and furans studied and oxidation conditions (using the PAM-OFR) with values reported in the literature (experimental conditions are specified).**

| SOA precursor | Experimental conditions | SOA yields (%) | References |
|---|---|---|---|
| Naphthalene | OH, PAM-OFR, RH = 41% | 61.5 - 69.9 | This study |
| | NO$_3$, PAM-OFR, RH = 43 % | 15.7 – 19.5 | This study |
| | OH, smog chamber, low NO$_x$, 5 < RH < 8%, seed | 73 | (Chan et al., 2009) |
| | OH, smog chamber, high NO$_x$, 5 < RH < 8%, seed | 19 - 30 | (Chan et al., 2009) |
| | OH, smog chamber, low NO$_x$, RH < 5% | 8 - 16 | (Shakya and Griffin, 2010) |
| | OH, smog chamber, low NO$_x$, RH < 3%, seed | 18 - 36 | (Kleindienst et al., 2012) |
| | OH, smog chamber, high NO$_x$, RH = 30%, seed | 11 - 29 | (Kleindienst et al., 2012) |
| | OH, smog chamber, low NO$_x$, RH < 0.1% | 4 - 31 | (Chen et al., 2016) |
| | OH, smog chamber, high NO$_x$, RH < 0.1% | 3 - 60 | (Chen et al., 2016) |
| | OH, smog chamber, low NO$_x$, RH < 0.1% | 21 - 50 | (Chen et al., 2018) |
| | OH, OFR, RH = 57% | 28 ± 6.7 | (Wang et al., 2018) |
| Acenaphthylene | OH, PAM-OFR, RH = 39% | 62.4 - 78.2 | This study |
| | NO$_3$, PAM-OFR, RH = 49% | 39.0 - 49.8 | This study |
| | OH, smog chamber, low NO$_x$, RH < 5% | 4 - 13 | (Shakya and Griffin, 2010) |
| | OH, smog chamber, low NO$_x$, RH < 1% | 61 | (Riva et al., 2017) |
| | OH, smog chamber, high NO$_x$, RH < 1% | 46 | (Riva et al., 2017) |
| Fluorene | OH, PAM-OFR, RH = 39% | 54.9 - 67.9 | This study |
| Phenanthrene | OH, PAM-OFR, RH = 27% | 59.1 - 88.1 | This study |
| | NO$_3$, PAM-OFR, RH = 43 % | 16.7 - 20.1 | This study |
| | OH, OFR, RH = 57% | 12 ± 2.6 | (Wang et al., 2018) |
| 2-Methylfuran | OH, PAM-OFR, RH = 38% | 12.3 - 12.5 | This study |
| | NO$_3$, PAM-OFR, RH = 49% | 3.3 - 3.5 | This study |
| | OH, smog chamber, high NO$_x$, dry conditions | 5.5 ± 1.6 | (Gómez Alvarez et al., 2009) |
| 2,5-Dimethyfuran | OH, PAM-OFR, RH = 38% | 9.0 - 9.8 | This study |
| | NO$_3$, PAM-OFR, RH = 53% | 4.0 - 4.4 | This study |
| | OH, smog chamber, 25 < RH < 60% | 0.4 - 6.2 | (Tajuelo et al., 2021) |
| | OH, smog chamber, high NO$_x$, RH < 10% | 0.9 - 1.2 | (Tajuelo et al., 2021) |
| Furan | OH, PAM-OFR, RH = 34% | 9.4 - 11.5 | This study |
| | OH, smog chamber, high NO$_x$, 5 < RH < 88%, seed | 0.04 - 5 | (Jiang et al., 2019b) |
| | OH, smog chamber, high NO$_x$, dry conditions | 1.9 - 7.2 | (Gómez Alvarez et al., 2009) |
| | NO$_3$, smog chamber, RH = 2 - 16 % | 7 | (Jiang et al., 2019a) |



### 3.4 SOA light absorption properties ($b_{abs}$, $\alpha$, and MAC)

Figure 6a shows the wavelength-dependent MAC values (from 370 to 590 nm) for SOA derived from the 4 PAHs and 3 furans
studied here with OH and $NO_3$ reactivity. A comparison with MAC values, from 300 to 550 nm, reported in the literature for laboratory-generated SOA from the oxidation of furan, monoaromatic and phenolic compounds (Lambe et al., 2013; Liu et al., 2016; Xie et al., 2017a; Jiang et al., 2019a) is also presented. MAC values alone together with $b_{abs}$ are shown in Figure S4. Overall, the shape of the spectra is characteristic of atmospheric BrC materials, with higher absorption in the UV range (Laskin et al., 2015; Hems et al., 2021; Liu et al., 2016; Siemens et al., 2022; Moise et al., 2015). Our results showed significantly
lower light absorption for furans SOA than PAHs SOA for both, day- and nighttime oxidation conditions. No light absorption was observed for furans SOA for wavelengths > 370 nm (Ångström exponents were then not calculated for furans SOA). At this wavelength of 370 nm, the highest MAC value was observed for Acy with OH radicals ($0.97\pm 0.06$ m$^2$ g$^{-1}$). Other PAHs SOA generated with OH radicals showed comparable MAC values in the range of 0.36 - 0.44 m$^2$ g$^{-1}$. Several studies have reported MAC values for Naph SOA, at 400-405 nm, in the range of 0.02-0.35 m$^2$ g$^{-1}$ with low $NO_x$ (Updyke et al., 2012;
Lambe et al., 2013; Lee et al., 2014; Xie et al., 2017a; Siemens et al., 2022; Metcalf et al., 2013; Klodt et al., 2023; He et al., 2022) (Table S7). Results at 400 nm, with an average MAC value of 0.22 m$^2$ g$^{-1}$, were in good agreement with the literature data obtained from filter extracts. The same applies to the absorption Ångström exponent ($\alpha$) for which the value of $5.43 \pm 0.15$ obtained was comparable to the literature data (ranging from 5.2 to 8.9). These results show that the filter tape-based optical method (aethalometer) applied here can provide a good estimate of such aerosol optical properties. However, using
AE33 data, the characterization of the UV region is quite limited at 370 nm while major light absorption for SOA has been mainly reported from 350 nm and below.

Over the PAH SOA formed with OH radicals, Naph showed the lowest $\alpha$ value (Figure S5). Acy SOA had the highest $\alpha$ of $8.35 \pm 0.25$, while Phe and Flu showed comparable values ($6.92 \pm 0.48$ and $6.08 \pm 0.51$, respectively). With $NO_3$, the $\alpha$ values for Naph ($5.57 \pm 0.12$) and Phe SOA ($7.05 \pm 0.88$) were similar as with OH exposure while a significantly lower $\alpha$ value was
obtained for Acy ($6.78 \pm 0.13$), and comparable to Phe. Similarly, the MAC value (at 370 nm) of Acy SOA formed with $NO_3$ radical ($0.41 \pm 0.12$ m$^2$ g$^{-1}$) was largely lower than with OH radical and comparable to Phe ($0.43 \pm 0.10$ m$^2$ g$^{-1}$) and, in a lesser extent, to Naph ($0.32 \pm 0.03$ m$^2$ g$^{-1}$). The MAC values for furans SOA were significantly lower and about 0.010 m$^2$ g$^{-1}$ for 2-MF, 2,5-DMF, and furan with OH radicals, and ranging from 0.005 to 0.021 m$^2$ g$^{-1}$ for 2-MF and 2,5-DMF with $NO_3$ radicals. A similar range has been reported for furan SOA formed by oxidation with $NO_3$ radicals (0.08-0.11 m$^2$ g$^{-1}$) (Jiang et al., 2019a;
Chen et al., 2022b). While furans SOA exhibits a low light absorption in the UV region, PAH SOA absorption properties are significant. Only cresol and benzene SOA formed under high $NO_x$ concentrations show higher MAC values in the UV-visible region (Figure 6a). Comparing MAC values determined here with those from ambient air biomass burning organic aerosols (BBOA) (Figure 6b) (Moise et al., 2015; Hoffer et al., 2006; Lack et al., 2012; Washenfelder et al., 2015; Zhang et al., 2016; Xie et al., 2017b; Cappa et al., 2019; Soleimanian et al., 2020; Mbengue et al., 2021), demonstrates that PAH SOA are a
significant BrC component of the emissions following biomass burning.





Finally, MAC values obtained for PAH and furans SOA with $NO_3$ radicals were similar, or even lower for Acy SOA, to values observed with OH radicals. Only the $NO_3$ radical with 2,5-DMF gave higher MAC values. Usually, higher light absorption values are observed for anthropogenic SOA formed under a high $NO_x$ regime due to the formation of stronger nitrogen-containing chromophores (nitroaromatic mainly) (Moise et al., 2015; Hems et al., 2021; He et al., 2022; Liu et al., 2016; Klodt

et al., 2023; Laskin et al., 2015; Siemens et al., 2022; Xie et al., 2017a). The SOA formation with $NO_3$ radical alone does not induce necessarily a large formation of nitro-organic species. For PAHs, a significant formation of nitro-chromophores species is usually observed upon heterogeneous oxidation processes with $NO_3$ radical (gas/particle) but not for homogenous reactions in the gaseous phase (Cheng et al., 2020; Kwamena and Abbatt, 2008; Lu et al., 2011; Keyte et al., 2013). Reactions involving $NO_3$ interaction with substituent groups through H-atom abstraction, for example for Ace, Acy, and Flu, are not expected to

induce an addition of $NO_2$ on the by-products formed (Keyte et al., 2013; Zhou and Wenger, 2013a, b). The reaction of Naph with $NO_3$ alone induces a formation of nitronaphthalenes and nitronaphthols that are mainly in the gaseous phase and are not associated with the SOA formed (Keyte et al., 2013).



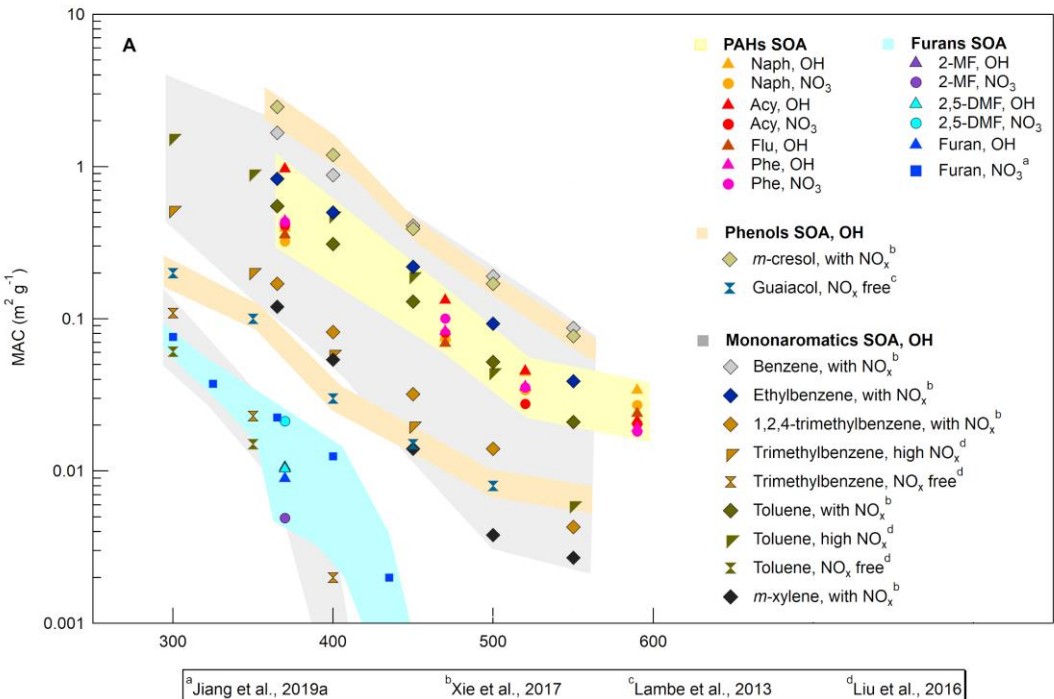

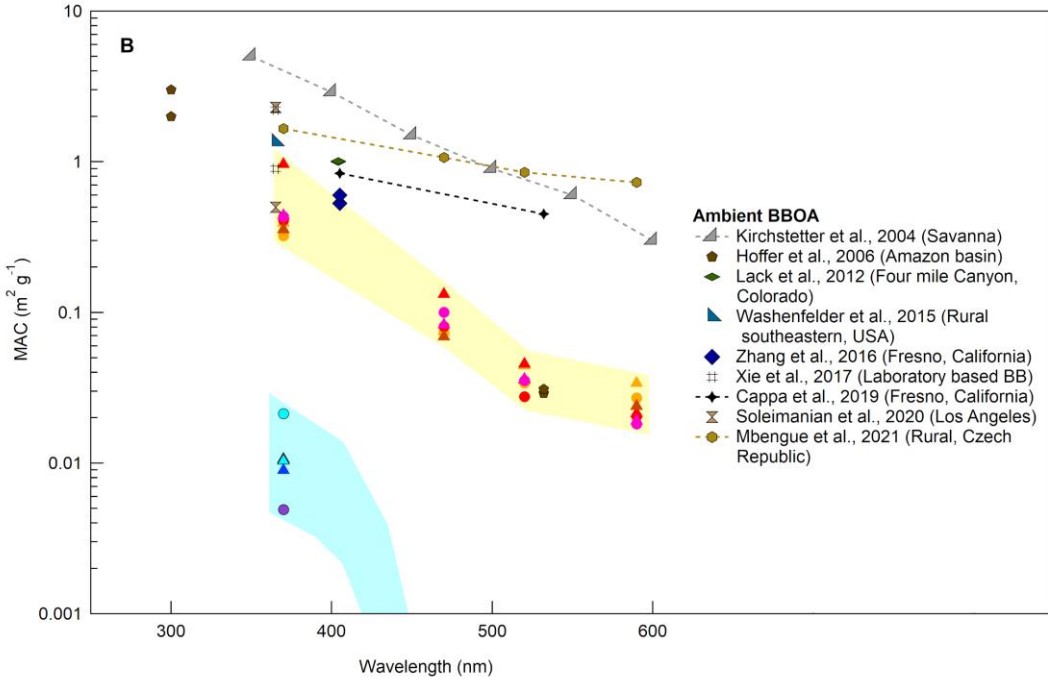

**Figure 6. Comparison of the mass absorption cross section (MAC) from the day- and nighttime oxidation processes (with OH and NO₃ radicals respectively) with literature data for laboratory-generated SOA from anthropogenic precursors (A) and with data for ambient air biomass burning organic aerosols (BBOA) (B).**



## 4 Conclusion

This study provides new key information on the properties of SOA formed from the oxidation of typical precursors emitted by
biomass burning, namely PAHs and furans, under both day- and nighttime chemistry. Overall, PAH SOA formation yields
were 5-6 times greater than those evaluated for furans (3 - 12%), illustrating the significant potential of PAHs in terms of SOA
formation. Under our conditions, without $NO_x$, the SOA yields observed with $NO_3$ radicals were 3 times lower than with OH.
The SOA formation yields obtained here from OFR experiments were either consistent with or higher than those reported in
previous studies, essentially from environmental (smog) chamber experiments. This might be the result of the different
experimental conditions applied in terms of oxidant, SOA precursor, $NO_x$ concentrations, and humidity. However, our results
showed again that OFR is a relevant alternative and complementary tool to smog chambers to simulate atmospheric aging
(Bruns et al., 2015; Peng and Jimenez, 2020). The SOA effective density showed an increasing trend with particle size for
both reactivities before reaching a "plateau", highlighting potential differences in terms of chemical composition and
morphology with the particle size, and thus a need to investigate them in future studies. PAHs SOA showed a higher light
absorption in the range of 370 to 590 nm by comparison to furans that do not show significant absorption at higher wavelengths
than 370 nm. A significant contribution of PAH SOA to ambient air BrC linked to biomass burning emissions is then expected.
Finally, no increase in the MAC values was observed from OH to $NO_3$ oxidation processes, probably due to a low formation
of nitrogen-containing chromophores through homogeneous gas phase oxidation processes with $NO_3$ only (without $NO_x$). As
it is well known that $NO_x$ has a significant impact on the SOA formation (in terms of yields for instance) and their optical
properties, with the subsequent formation of strong absorbing chromophores, experiments performed under high $NO_x$
conditions, with both OH and $NO_3$ radicals, will be the focus of future works.

## Competing interests

The authors declare that they have no conflict of interest.

## Author contribution

AA designed and led the research. AEM and AA performed the experiments. AEM and AA analyzed the data. AEM, AA, BD,
ZP, OF, JEP, LD and ATL contributed to the interpretation of the data and results. AA was responsible for funding acquisition.
AA, SAA and BD supervised AEM PhD work. AEM and AA wrote the manuscript with inputs from all co-authors.



**Acknowledgments**

This work has been supported by the French Ministry of the Environment. The authors gratefully acknowledge Serguei
Stavrovski, Ahmad El Masri, Robin Aujay, Nicolas Karoski, Tanguy Amodeo and Laurent Meunier (Ineris) for their help on
the different instrumentations and for sample preparation and PAH analyses.

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
