# Peer review of "Insights into secondary organic aerosol formation from the day- and nighttime oxidation of PAHs and furans in an oxidation flow reactor"

_EGUsphere, 2023_

## Author Comment (AC1)

Verneuil en Halatte, 26th September 2023

Dear Arthur Chan,

Please find below a point-by-point response to the referees' comments (in blue) concerning the manuscript EGUSPHERE-2023-1355 entitled "Insights into secondary organic aerosol formation from the day- and nighttime oxidation of PAHs and furans in an oxidation flow reactor". We have addressed each of the reviewer's comments and revised the manuscript accordingly. We think that this new version can now fully meet the standards of the Atmospheric Chemistry and Physics journal.

With very best wishes,

Dr. (HDR) Alexandre ALBINET

INERIS

Parc Technologique ALATA – BP 2

Verneuil-en-Halatte, 60550 (France)

Email:        alexandre.albinet@gmail.com; alexandre.albinet@ineris.fr

First, we would like to thank the four anonymous referees for their constructive comments on our work and this paper. All comments have been considered and the manuscript is now improved. Changes made are highlighted in yellow in the marked version of the revised manuscript and supplementary information.

**Referee #1:**

The authors presented experimental results of secondary organic aerosols (SOA) formation from oxidation of three furans and four polycyclic aromatic hydrocarbons (PAHs) in an oxidative flow reactor (OFR). Experiments were conducted with OH and $NO_3$ oxidation, representing day- and night-time chemistry. Results include SOA yields, effective density, absorption Angstrom exponent, and mass absorption coefficient. The authors concluded from the results that PAHs had higher SOA yields and PAH SOA had higher absorption capability compared to furans. A particularly interesting finding is that SOA density generally increased with particle size. Another important result is that $NO_3$ oxidation of these precursors did not result in higher absorption compared to OH oxidation, because $NO_2$ might be needed to form light-absorbing species. The experiments were well planned and conducted, and the manuscript is well written. There are, however, a few things I would suggest the authors to clarify. I therefore recommend Major Revision, with comments shown below.

Main:
1. My main concern for this study is that the precursor concentrations were super high (mg per cubic meter level), resulting in super high SOA loading (hundreds or even over 1000 of microgram per cubic meter). The authors might need to better justify this because it is quite far away from ambient conditions. There are some previous studies showing that SOA properties (including mass yield) might be loading dependent. One particularly important aspect is that higher loadings might favor partitioning of volatile (and less oxygenated) species into the particle phase, thus affecting the density measured.

We agree with the reviewer's concern. The high concentrations of SOA generated may have an impact their physicochemical properties. However, we can finally observe that the results obtained

here, even with such concentrations, were in good agreement with previous studies in terms of light absorption properties (MAC values), effective densities (see below and Figure A1 for additional explanation) and, in a lesser extent, SOA yields (even if the results obtained were in the upper range probably due to the high loadings too). In addition, such high concentrations in terms of PM may reflect the conditions encountered in biomass burning plumes due to forest fires for instance. Besides the study of the SOA formation from the oxidation of PAHs and furans, our objective was also to collect enough SOA quantity (on filters) to perform further *in vitro* biological assessment. and investigate the SOA biological activities (El Mais, 2023). We have thus purposefully increased the SOA precursor concentrations to generate high SOA loadings. It was a matter of compromise between both objectives namely, SOA physicochemical characterization and *in vitro* bio-assessment.

This was explained in the first version of the manuscript and the sentence has been modified to better clarify both objectives (Lines 145-148): "These concentrations were relatively high and may represent concentrations encountered in large forest fire plumes. Besides the study of the SOA formation, our objective was to produce and collect enough quantities of PM (collected on filters, sampling flow of 6.5 L min$^{-1}$, Figure 1) to further perform in vitro biological assessment (El Mais, 2023)."

2. If the authors want to compare the SOA from seven precursors, how come the level of oxidants and equivalent aging time were so much different among experiments (Tables 1 and 2). External OH reactivity might affect those conditions. But it is not very straightforward to compare SOA for different precursors as well as different aging extents.

The radical exposure was also dependent on the precursor concentrations injected and the SOA mass produced which were specific for each precursor and oxidant. As specified before, we have also adjusted in some cases the lamps voltages (between 1.8 and 3 V) to generate higher SOA concentrations.

3. The finding that SOA density increased with particle size is interesting. But it is a little bit counterintuitive. Let's say, smaller particles are dominated by very non-volatile species that can potentially nucleate, and easily condense (partition) into the particles because of the

low volatility; those species should be very oxygenated and have high density. Large particles, on the other hand, are formed from condensation of lots of less volatile species, which should be less oxygenated, thus possess lower density. The net result, from a wild guess, would be a decreasing trend of density as SOA particle size increases. While my wild guess might not be true, I would suggest the authors elaborate a bit more than what is shown in P12.

Overall, SOA density increasing trend observed was also surprising for us and it seems interesting for all referees. The assumption made by the reviewer may be true with respect to the chemical composition, but the observations made showed something different. To support the discussion here, we have also studied during other experiments the secondary particle formation by nucleation processes (meaning reactivity only from gaseous phase emissions as here) from logwood stove (LWS) emissions. LWS emissions entering the PAM-OFR were filtered using a radial filter (being aware of probable loss of some IVOCs). Note that, the concentrations of the SOA particles into the PAM-OFR where about 3 to 5 times lower than in the PAH and furan SOA experiments. An example of results obtained in nominal output (100 % of the power loading of the LWS) is presented on Figure A1. The effective density observed increased from 1.0 to 1.45 g cm$^{-3}$, with the particle size in the range of 30 to 200 nm before reaching a "plateau" around 1.4 g cm$^{-3}$. Our findings were consistent with those obtained for PAH and furans SOA. Such results contrasted with the observations reported recently for aged particle emissions from biomass burning or brown coal (including then soot particles) where a decreasing trend, or rather stable, was observed as well as for primary particle emissions (Leskinen et al., 2023). We also observed the same when performing similar experiments and we also noticed the increase in density between primary and aged emissions.

Different parameters may have a role on such behavior in our study including the humidity conditions, the concentrations of particles but so far, we do not have any clear explanation of the results obtained here for such SOA formed by nucleation. More research should be done in to study the potential differences in terms of chemical composition and morphology with the particle size. This will be the purpose of future studies.

[Figure]

**Figure A1.** Effective density evolution of the logwood stove (LWS) SOA formed by nucleation. Results obtained in nominal output (100% of the LWS power loading) with OH radicals. Results are presented as a function of the aerosol mobility diameter. The error bars correspond to the standard deviations (2 σ) on the number of measurements replicates (n= 2 to 4).

Technical:

1. P3/L63: ozone is not a radical?

The reviewer is right. Ozone is an oxidant but not a free radical. Modification has been made.

2. P3/L74: if the OA components are non-absorbing, they will not be classified as BrC, right?

This part of the sentence has been removed.

3. P7/L2: just "blanks"? It is not a field study.

Replaced by "experimental blanks".

4. P8/L205: a unit of either molecules per cubic centimeter or ppt for NO3. Otherwise, it is not straightforward to make a direct comparison.

The unit is now specified in molecules cm$^{-3}$.

5.  P15/L318: Joo et al. (2019a)? There are a few other places that Author (Year) format should be used.

This has been modified in all the text.

6.  Did the authors have $NO_2$ concentration data from either KinSim model or measurements to rule out the involvement of $NO_2$ to form light-absorbing products?

$NO_2$ concentrations were below the limit of detection (= 1 pppm) of the $NO_x$ analyzer used. This is now specified in the text. Lines 134-135: "In such conditions, $NO_2$ concentrations in the PAM-OFR ($NO_X$ Analyzer 42i-HL, Thermo Scientific) were below the quantification limit of 1 ppm (in the range used of 0 - 200 ppm)". Additionally, the same results were obtained from the KinSim simulations (Table A1). Resulting $VOC_0/NO_2$ agreed with low $NO_x$ regime conditions ($VOC/NO_x > 10$) (Zhang et al., 2023; Srivastava et al., 2022a). Only for phenanthrene this ratio was low and similar to high $NO_x$ regime conditions ($VOC/NO_x < 3$). Overall, both, measurements, and modelling results, suggested that no $NO_2$ (or very low concentrations) was available into the PAM-OFR and involved in the formation of light-absorbing properties.

**Table A1.** $NO_2$ concentrations obtained from KinSim modulations.

| SOA precursor | $NO_2$ concentrations | | $VOC_0/NO_2$ ratio |
| --- | --- | --- | --- |
| | molecules cm$^{-3}$ | ppm | |
| 2-MF | $2.47 \times 10^{13}$ | 1.01 | 12.4 |
| 2,5-DMF | $3.41 \times 10^{13}$ | 1.39 | 9.2 |
| Naph | $1.89 \times 10^{12}$ | 0.08 | 36.1 |
| Acy | $8.15 \times 10^{12}$ | 0.03 | 16.3 |
| Phe | $2.00 \times 10^{12}$ | 0.08 | 0.5 |

This manuscript describes studies of SOA formation by PAH and furan compounds present in smoke plumes by long-term oxidation with OH and $NO_3$ radicals in an oxidation flow reactor. The density, absorption, and yields of the SOA formed are probed by a variety of instruments. The authors find SOA yields and absorption to be much lower for $NO_3$ radical reactions than OH radical reactions, and also for furan reactions than PAH reactions. They also find that density depends on particle diameter for particles below 100 nm, an especially interesting result. The modeling of direct photolysis of precursors, and its comparison to the OH and $NO_3$ reactions being studied, is very helpful. For many of the precursors studied, this work is the first one to measure SOA yields using $NO_3$ radicals. This work will clearly be of interest to atmospheric chemists but needs minor revision to address the comments below.

Overall, the introduction is very clear. The introduction would be enhanced, however, by some discussion of the emission levels of PAHs and furans in biomass burning plumes to establish the environmental significance of this project. I consulted the reference{Oros, 2001 #2951} which quantified substantial PAH emissions from pine wood burning, but furan emissions were not reported.

This information has been added with the following, Lines 54-56: "Aromatic hydrocarbons (including PAHs), oxygenated aromatic (including phenolic species) and furans constitute a significant fraction of the total VOCs emitted by biomass burning (10 – 50 % (with 10 - 20 % of PAHs), 5 - 20% and 5 - 40%, respectively) (Hatch et al., 2015; Schauer et al., 2001; Bruns et al., 2016; Bhattu et al., 2019)."

The idea that differences in morphology might explain the dependence of particle density on particle diameter seems implausible without some further rationalization. While the authors are careful to use words like "possibly" and "probably" for this claim (please note that the meaning of these two words is quite different!), it seems much less likely to explain the observed aerosol densities than differences in chemical composition, since the density rises with diameter: this is the opposite of what one might expect for particles formed by agglomeration of smaller, solid particles. Furthermore, if the particles are liquid, a non-spherical shape is unlikely. Some way of

rationalizing how particle morphology might cause the observed density trend is needed in the discussion. Citing other studies of size-dependent SOA properties, if the authors know of any, could also be helpful.

There is no general pattern relating the effective densities and PM size. This is rather dependent on the source, chemical composition, mixing state, morphology, and aging conditions (Li et al., 2016). We did not find any previous work showing the size-dependent density of SOA particles formed by nucleation but only from aged emissions (so including soot particles) (Leskinen et al., 2023). In the latter case, a decreasing trend, or rather stable, was reported, as well as for primary particle emissions (Leskinen et al., 2023). For information, we also observed the same when performing similar experiments from logwood stove (LWS) emissions. and we also noticed the increase in density between primary and aged emissions. As specified before (Reviewer 1), the secondary particle formed by nucleation processes from LWS emissions (reactivity only from gaseous phase emissions by filtering the emissions before entering the PAM-OFR) showed the same trend as here for PAH and furan SOA formed by nucleation (Figure A1) supporting our findings here.

We have modified the text and now use the terms "possibly" when explaining that the results observed might be related to some differences in terms of morphology of the particles. We are aware that such results will require deeper investigations in the future including the sampling of the SOA formed on TEM grids to have a look on the shape of the particles generated in such conditions.

The authors conclude that PAH SOA has significant UV-vis absorption, but also note that MAC does not increase when OH or NO$_3$ exposures are increased. How do these two claims relate to each other? They might seem contradictory to readers without further explanation.

We think there is some sort of misunderstanding here. We did not report that "MAC does not increase when OH or NO$_3$ exposures are increased" because we did not study the MAC as a function of OH or NO$_3$ exposures. However, we noticed that "MAC does not increase from OH to NO$_3$ oxidation processes" and we attributed that to the probable low formation of nitrogen-containing chromophores with NO$_3$ only (without NO$_2$).

Section 2.3: The verbal description of the aerosol sampling instruments is inconsistent with Figure 1. Specifically, the use of terms "in series" and "in parallel" in this section is very confusing. Based on Figure 1, the only instruments that are sampling in series are the density instruments (DMA + CPMA + CPC) in the bottom row. The instruments in the middle row are sharing a common sampling line, but they are sampling from it in parallel.

Corrected accordingly.

Line 201: It seems problematic to run an OFR experiment where the precursor is completely destroyed before exiting the reactor, and the oxidant concentrations therefore rise rapidly. Why was this done?

The KinSim simulations have been performed afterwards i.e. after the experiments were completed. Such calculations are usually not even performed in the studies using OFR. We can finally see how they are beneficial in order to discuss the results obtained. From now on, and as we are aware of such process into the PAM-OFR, we first try to simulate the behavior of the precursors and oxidants using KinSim that will help us in the design of the experiments and improve them.

Equation 4: The discussion of this equation centers on how the multiple scattering correction needed to be raised beyond the one provided by the instrument manufacturer. While this is not a concern for aethalometry – it is standard practice – equation 4 does not contain this term, and so it is unclear from the presentation how the larger C value used in this work influences the results.

Equation 4 has been modified to show the C value provided by the manufacturer and the new C value as measured by Drinovec et al. (2022). Lines 233-234:

"$b_{abs}(\lambda) = BC\,(\lambda) \times \sigma_{air}(\lambda) \times \frac{C_{Manufacturer}}{C_{new}}$"

Line 238: The corrected absorption is smaller by the factor $C_{new}/C_{Manufacturer}$.

Figure 3: It is unclear which vertical axis goes with the furan + OH oxidation size distribution data. Is it shared with 2,5-DMF or with the other five compounds (the leftmost axis)?

The figure has been modified to make it clearer. 5 compounds share a first common Y-axis (on the left) and 2,5-DMF and furan share the second Y-axis.

[Figure]

Section 3.2: This discussion omits any mention of the 10x larger numbers of aerosol particles produced by OH oxidation compared to NO3 oxidation for four of the precursors. It makes sense that the aerosol particles from NO3 oxidation grow substantially larger, since there are fewer of them available for gas-phase product molecules to condense onto (or dissolve into). For 2,5-DMF, where the aerosol size trend seems the opposite of the other precursors (OH oxidation produces larger particles), it can also be explained by an opposite trend in aerosol numbers (OH oxidation produced fewer particles).

The text has been modified to integrate such discussion (Lines 272-276): "Moreover, the number of aerosol particles produced from Naph, Acy, Phe, and 2-MF by OH oxidation was 10 - 100 times larger compared to $NO_3$ oxidation. The aerosol particles from $NO_3$ oxidation grew substantially larger since fewer of them were available for semi-volatiles species produced to condense onto. However, for 2,5-DMF, the particle size distribution shifted towards smaller particles ($D_p \sim 35$ nm) even though the concentration injected was about 3 times higher than during OH exposure studies.

This might be also linked to the lower particle concentrations in the system observed with OH oxidation".

Table 3: It would be helpful to compare the densities to those of the precursor species. Could the variability in precursor densities help to explain the variability in SOA density?

Table A2 compares the densities of the SOA to their corresponding precursors. The variabilities observed in the SOA densities does not seem linked to the density of their precursors. We did not include such information in the text or in Table 3 because it does not seem relevant.

**Table A2.** Comparison of the densities between the PAHs and furans pure precursors and their corresponding SOA formed under day and nighttime conditions (OH and $NO_3$ radicals respectively). SOA densities were obtained from the mean of the densities corresponding to SOA with $d_m > 100$ nm.

| Precursor | Precursor density (g cm$^{-3}$) | Aerosol effective densities (g cm$^{-3}$) | |
| --- | --- | --- | --- |
| | | OH radical | NO$_3$ radical |
| Naphthalene | 1.14 | 1.33 ± 0.11 | 1.39 ± 0.06 |
| Acenaphthylene | 0.90 | 1.23 ± 0.05 | 1.47 ± 0.13 |
| Fluorene | 1.2 | 1.60 ± 0.10 | No SOA formation |
| Phenanthrene | 1.18 | 1.52 ± 0.02 | 1.29 ± 0.11 |
| 2-methylfuran | 0.93 | 1.52 ± 0.06 | 1.38 ± 0.06 |
| 2,5-dimethylfuran | 0.89 | 1.35 ± 0.08 | 1.35 ± 0.08 |
| Furan | 0.94 | 1.61 ± 0.05 | Not determined |

Line 365: this claim appears to be a logical error or at least an oversimplification. Contribution to biomass burning BrC is also determined by the level of emissions of the precursor gases – if the emissions are small, they may not contribute significantly to biomass burning BrC even if they produce highly absorbing SOA in single-precursor studies like these. This is another reason to reference emission studies in the introduction section. In addition, POA absorption in biomass burning plumes is significant. The absorbance of a single-precursor oxidation SOA cannot be

directly compared to the absorbance of biomass burning aerosol without taking these other issues into account.

This a good point and it is now first elucidated with the information provided in the introduction about the contributions of key SOA precursors (monoaromatics, oxygenated aromatics, furans, and PAHs) to the total VOC emissions from biomass burning. In addition to the emission level, SOA formation and SOA absorption properties need to be considered. We have now specified in the text the following (Lines 379-385): "PAHs emissions from biomass burning (except naphthalene which is in the same range) are about 4 to 10 times lower than benzene, toluene, phenol or furans and comparable to other monoaromatics and phenolic species (Bruns et al., 2016; Schauer et al., 2001; Bhattu et al., 2019; Hatch et al., 2015). Considering the SOA formation yields of these species and their contribution to the total SOA produced from biomass burning emissions (Bruns et al., 2016) (e.g comparable for naphthalene and benzene, twice lower for phenol and 4 times higher for m-cresol) and considering the light absorption properties of the different SOA (Figure 6), we can figure out that PAH SOA are a significant BrC component of the emissions following biomass burning".

Line 372: the reasoning here is confusing. Once aerosol particles have been formed, couldn't these experiments be largely observing heterogeneous oxidation processes, even though initially gas-phase reactions are the only thing happening?

This is right, heterogeneous reactions may also take place inside the OFR. The sentence has been modified as follows to be clearer (Lines 391-396): "For PAHs, a significant formation of nitro-chromophores species is usually observed upon heterogeneous oxidation processes with $NO_3$ radical (gas/particle) but not for homogenous reactions in the gaseous phase (Cheng et al., 2020; Kwamena and Abbatt, 2008; Lu et al., 2011; Keyte et al., 2013). Once the aerosols formed in the PAM-OFR, heterogenous reactions may occur but reactions involving $NO_3$ interaction with substituent groups through H-atom abstraction, for example for Ace, Acy, and Flu, are not expected to induce an addition of $NO_2$ on the by-products formed (Keyte et al., 2013; Zhou and Wenger, 2013a, b)".

Technical corrections

Line 350:  "quite limited at 370 nm" could be more clearly expressed as "limited to 370 nm"

Corrected accordingly.

**Referee #3:**

Abd El Rahman El Mais et al. investigated secondary organic aerosol (SOA) formation from the oxidation of furanoids and polycyclic aromatic hydrocarbons (PAHs) using an oxidation flow reactor (OFR) with two different oxidants: OH and $NO_3$ radical. The authors explored the size-dependent effective density of SOA, SOA yield, and SOA light absorption and found that PAH SOA shows higher SOA yield and stronger light absorption than furanoid SOA. Understanding these parameters from such less-studied volatile precursor compounds would improve our understanding of biomass burning impact on air quality and climate change. Overall, the manuscript is well-written and well-displayed. However, I have major comments that need to be addressed before publication.

1. The quantitative analysis requires a more thorough evaluation. Authors are reporting SOA yield, but the wall loss effect is evaluated based on the size range where the mode of number concentration is observed, instead of that of the volume concentration. Authors should diagnose and report if the particle loss at the size range of volume concentration mode was also little as what they have observed from the number concentration loss. Also, SOA yield is a function of organic mass concentration formed during oxidation. The comparison among precursors or with previous studies should consider the organic mass concentration comparison as well. Lastly, authors should address the reason why they studied under atmospheric irrelevant conditions (i.e., high precursor VOCs concentration, no-NOx condition, absence of pre-existing particle).

   - The particle loss using ammonium sulfate aerosols in the range of 30 - 200 nm, was check in terms of particle number and volume concentrations and they were for both below 5% for all PM sizes.

   - The information regarding the organic (PM) mass concentrations from literature data and for our experiments have been added to the Table 4 in the main text. The discussion regarding the parameters influencing the results obtained in terms of SOA yields has been also updated in this way (Lines 331-338): "Several other factors have an influence the SOA yields such as the $NO_x$ concentrations (or ratio VOC/$NO_x$), the use and type

of seed particles, PM (organic) mass concentrations, temperature and RH conditions (Srivastava et al., 2022b; Lambe et al., 2015; Zhang et al., 2023). It has also been shown recently that smog chamber and OFR studies probably overestimate SOA formation yields compared to atmospheric conditions due to high peroxy radical ($RO_2$) concentrations leading to overemphasis of cross reactions of $RO_2$ compared to the reaction with $HO_2$ (or $NO_3$) (Schervish and Donahue, 2021). Thus, the high VOC and PM (OA) concentrations together with RH% and $RO_2 + RO_2$, $RO_2 + HO_2$ or $RO_2 + NO_3$ reaction processes, may have enhanced the SOA formation here explaining the differences observed with literature data (Table 4)".

- As explained above to Reviewer 1, besides the study of the SOA formation from the oxidation of PAHs and furans, our objective was also to collect enough SOA quantity (on filters) to perform further in vitro biological assessment. and investigate the SOA biological activities (El Mais, 2023). We have thus purposefully increased the SOA precursor concentrations to generate high SOA loadings. It was a matter of compromise between both objectives namely, SOA physicochemical characterization and in vitro bio-assessment. This was explained in the first version of the manuscript and the sentence has been modified to better clarify both objectives (lines 145-148): "These concentrations were relatively high and may represent concentrations encountered in large forest fire plumes. Besides the study of the SOA formation, our objective was to produce and collect enough quantity of PM (collected on filters, sampling flow of 6.5 L min$^{-1}$, Figure 1) to further perform *in vitro* biological assessment (El Mais, 2023)".

Finally, as for many other studies (Srivastava et al., 2022b; Zhang et al., 2023), it is not possible to cover all the conditions that actually exist in the atmosphere in a single study and we had to make some choices. So we decided first to focus on the humidity to have something environmentally relevant in that case (which is not the most usual in other studies especially for $NO_3$ reactivity usually performed in dry conditions) and not to add $NO_x$ or seed particles. We plan to improve the experiments to now include different parameters in terms of $NO_x$ and VOC concentrations, seeds, etc..

2. Precursor VOCs concentration varies depending on the experiment while the level of oxidant injection is relatively consistent. This is affecting the VOC reaction with the oxidants: a substantial fraction of 2-methlyfuran and 2,5-methylfuran reacting with O3. Such reaction conditions would affect the RO2 reaction channel (RO2+RO2, RO2+HO2, RO2+NO3), inducing each experiment to have respective SOA-forming RO2 reaction conditions. Thus, authors need to discuss thoroughly how the SOA yield and other parameters are different among the precursors and from previous studies depending on the VOC-to-NOx ratio or VOC-to-oxidant ratio (also, the presence of seed aerosol).

This is a good point and the discussion include now such information: (Lines 331-338): "Several other factors have an influence the SOA yields such as the NOx concentrations (or ratio VOC/NO$_x$), the use and type of seed particles, PM (organic) mass concentrations, temperature and RH conditions (Srivastava et al., 2022b; Lambe et al., 2015; Zhang et al., 2023). It has also been shown recently that smog chamber and OFR studies probably overestimate SOA formation yields compared to atmospheric conditions due to high peroxy radical (RO$_2$) concentrations leading to overemphasis of cross reactions of RO$_2$ compared to the reaction with HO$_2$ (or NO$_3$) (Schervish and Donahue, 2021). Thus, the high VOC and PM (OA) concentrations together with RH% and RO$_2$ + RO$_2$, RO$_2$ + HO$_2$ or RO$_2$ + NO$_3$ reaction processes, may have enhanced the SOA formation here explaining the differences observed with literature data (Table 4)".

3. The formation of light-absorbing compounds in SOA is generally associated with nitrogen-containing organics, such as nitro-aromatics, amines, imine, etc. However, experiments here are performed under the NOx-free conditions for OH radical reactions. It would be informative if the authors can provide a more detailed discussion on which composition (or chemical functionalities) of SOA could potentially contribute to the light absorption of SOAs and how such light-absorbing compounds are formed during the oxidation of PAHs & furanoids. In addition, MAC values indeed seem high, reaching the level of ambient biomass burning studies, but the Absorption Angstrom Exponent seems low compared to biomass burning studies. The authors should add a discussion on this as well.

We did not perform any detailed chemical analyses using the filter collected or using on-line instrumentation (we had only a Q-ACSM at that time); So, we do not have any clue about the chemical composition of the formed SOA and so about the potential light-absorbing compounds. This will be the purpose of future studies in which a detailed physicochemical characterization of the gaseous and particulate phases will be performed using advanced instrumentation such as a CI-ToF-MS (chemical ionization-time of flight - mass spectrometry), combining several ionization modes ($H_3O^+$, $NH_4^+$, $O_2^+$, $I^-$) and aerosol inlets [e.g., EESI, extractive electrospray ionization (EESI)].

The Absorption Angstrom Exponents obtained here are comparable to literature data for instance for naphthalene SOA (Table S7). The values are not low compared to biomass burning studies, they are largely higher. In the literature $\alpha$ values for biomass burning emissions containing both BrC and BC are usually about 2 (Helin et al., 2021; Zotter et al., 2017; Lindberg et al., 2022). An Absorption Angstrom Exponent value (370-660 nm) of 5.6 for wood burning SOA, which is comparable to our measurements with naphthalene and lower than other tested compounds, has been also reported previously (Kumar et al., 2018).

Technical comments:

Line 80: Since the authors use the symbol $\alpha$ for the absorption angstrom exponent, the acronym "AAE" seems unnecessary.

Modified and the acronym "AAE" was removed.

Line 293: It would be better to comment that using the density obtained from individual experiments can also reduce potential biases for quantitative analysis.

This information has been added to the text.

Line 322: Please clarify the conditions: e.g., presence of NOx (or [NOx]), type (or concentration) of seed particles.

Modified accordingly.

Line 349: brown carbon comparison between AE 33 & filter-extracted measurements can be relocated after describing absorption of SOA generated via NO3 radical reactions as both OH and NO3 radical experiments show good agreement with literature values.

We would prefer to keep this comparison in its place as it concerns only naphthalene SOA.

Line 386, Table 4, etc.: please keep consistency when referring to the range throughout the manuscript. It is mixed up with various formats (e.g., 5-6 times, 5 – 6 times, or 5 - 6 times).

Corrected accordingly. The format "5 - 6 times" was used in all the text.

The authors Mais et al. describe a series of OFR experiments oxidizing PAH and furan compounds and characterize the density and optical absorption of the formed SOA. The manuscript is well-written, generally clear, and thoroughly references prior literature. The parent VOCs and questions investigated in this work are important and of general interest to the atmospheric community. I believe the manuscript will be suitable for publication after the major and minor comments below are addressed.

Major comment

I believe more discussion needs to be added on the extent to which the generated SOA is representative of SOA that's likely to form in the atmosphere.

- The precursor concentrations used in this work are quite high compared to what's often used in laboratory work and observed during field measurements. High precursor concentrations have been tied to increased SOA formation, an increase in measured SOA yield, and changes in aerosol composition (via AMS f44/f43 fractions) in prior OFR experiments (Kang et al., 2011). It seems possible that the SOA yields reported in this work are generally higher than in past works because of this phenomenon. Could associated SOA compositional changes (for example, increased condensation of SVOCs) bias the measured particle density or absorption properties compared to typical ambient SOA?

  - The high concentrations of SOA generated may have an impact their physicochemical properties. However, we can finally observe that the results obtained here, even with such concentrations, were in good agreement with previous studies in terms of light absorption properties (MAC values), effective densities and, in a lesser extent, SOA yields (even if the results obtained were in the upper range probably due to the high loadings too). In addition, such high concentrations in terms of PM may reflect the conditions encountered in biomass burning plumes due to forest fires for instance. As specified to Reviewers 1 and 3; besides the study of the SOA formation from the oxidation of PAHs and furans, our objective was also to collect enough SOA quantity (on filters) to perform further in vitro biological assessment. and investigate the SOA biological activities (El Mais, 2023). We have thus purposefully increased the SOA

precursor concentrations to generate high SOA loadings. It was a matter of compromise between both objectives namely, SOA physicochemical characterization and in vitro bio-assessment. This was explained in the first version of the manuscript and the sentence has been modified to better clarify both objectives (lines 145-148): "These concentrations were relatively high and may represent concentrations encountered in large forest fire plumes. Besides the study of the SOA formation, our objective was to produce and collect enough quantity of PM (collected on filters, sampling flow of 6.5 L min$^{-1}$, Figure 1) to further perform in vitro biological assessment (El Mais, 2023)".

- This high precursor and high PM particle concentration probably favors the SOA generation and consequently might raise the SOA yields. The SOA yields are higher than the previously reported works due to humid conditions (RH = 25 - 62%) used here, as well as the high precursor concentrations injected and the high SOA loadings (1000 - 1500 µg m$^{-3}$). This is now clearly specified in the text (Lines 331-338): "Several other factors have an influence the SOA yields such as the $NO_x$ concentrations (or ratio VOC/$NO_x$), the use and type of seed particles, PM (organic) mass concentrations, temperature and RH conditions (Srivastava et al., 2022b; Lambe et al., 2015; Zhang et al., 2023). It has also been shown recently that smog chamber and OFR studies probably overestimate SOA formation yields compared to atmospheric conditions due to high peroxy radical ($RO_2$) concentrations leading to overemphasis of cross reactions of $RO_2$ compared to the reaction with $HO_2$ (or $NO_3$) (Schervish and Donahue, 2021). Thus, the high VOC and PM (OA) concentrations together with RH% and $RO_2 + RO_2$, $RO_2 + HO_2$ or $RO_2 + NO_3$ reaction processes, may have enhanced the SOA formation here explaining the differences observed with literature data (Table 4)".

- Although the reviewer is right that the associated compositional changes might change the assessed properties, we believe that this probability is small. As specified before (Reviewers 1 and 2), further evidence have been obtained by studying the secondary particle formed by nucleation processes from logwood stove (LWS) emissions (reactivity only from gaseous phase emissions by filtering the emissions before entering the PAM-OFR). Results showed the same trend as here for PAH and furan SOA formed by nucleation (Figure A1) supporting our findings here.

- The authors state that the NO2 concentration in the OFR was below their detection limit (line 133). What was the detection limit? Are such low levels typical of what's expected in regions where NO3 radical is expected to form at night? The authors briefly reference that NO2 addition to PAHs during oxidation can form chromophores (~line 371), and I ask that the authors discuss whether the lack of NO2 during NO3 radical oxidation in the present work leads to SOA that would likely have different absorption properties compared to what would likely form from NO3 radical oxidation in an ambient environment.

  - NO$_2$ concentrations were below the limit of quantification (= 1 ppm) of the NO$_x$ analyzer used. This is now specified in the text. Lines 134-135: "In such conditions, NO$_2$ concentrations in the PAM-OFR (NO$_X$ Analyzer 42i-HL, Thermo Scientific) were below the quantification limit of 1 ppm (in the range used of 0 - 200 ppm)". Additionally, the same results were obtained from the KinSim simulations (Table A1). Resulting VOC$_0$/NO$_2$ agreed with low NO$_x$ regime conditions (VOC/NO$_x$ > 10) (Zhang et al., 2023; Srivastava et al., 2022a). Only for phenanthrene this ratio was low and similar to high NOx regime conditions (VOC/NO$_x$ < 3). Overall, both, measurements, and modelling results, suggested that no NO$_2$ (or very low concentrations) was available into the PAM-OFR and involved in the formation of light-absorbing properties.
  - Although NO/NO$_2$ are primarily emitted by traffic emissions rather than biomass burning, the presence of NO$_2$ during NO$_3$ radical oxidation may have an impact on the SOA formed in terms of chemical composition resulting in different light absorption properties. The study of the influence of NO$_x$ will be the purpose of our future studies to investigate the impacts on SOA yields, density, chemical composition and light absorption properties.

- Line 276: can the authors cite any prior literature that observed size-dependent SOA composition differences during oxidation/formation within the same chemical system? Does the observed SOA density size dependence have any implications for SOA density when formed in a system with seed aerosol (such as a typical ambient environment) as opposed to the present work where particles nucleate?

We did not find any previous work showing the size-dependent density of SOA particles formed by nucleation but only from aged emissions (so including soot particles) (Leskinen et al., 2023). In the latter case, a decreasing trend, or rather stable, was reported, as well as for primary particle emissions (Leskinen et al., 2023). For information, we also observed the same when performing similar experiments from logwood stove (LWS) emissions. and we also noticed the increase in density between primary and aged emissions. As specified above and before (Reviewers 1 and 2), and in agreement with our results here for PAH and furan SOA formed by nucleation, further evidence have been obtained by studying the secondary particle formed by nucleation processes from LWS emissions (reactivity only from gaseous phase emissions) (Figure A1). The study of the SOA chemical composition according to the PM size will be the purpose of future work.

Line 195: the usage of the KinSim model is appropriate, but I believe either the results need to be discussed further or the model re-run. The authors observe both OH and NO3 concentrations to sharply increase after precursor VOC is consumed and state that this leads to high estimated exposures. However, in the actual OFR experiments, the initial VOC oxidation will produce first- and later-generation products that will also be available to react with generated radicals, providing additional reaction partners while also delaying consumption of the precursor VOC. This would, presumably, influence the time-dependent estimation of radical concentrations and overall radical exposure as well as estimates of the precursor consumed.

We agree with the reviewer's opinion and the KinSim model is not elaborated enough to do such detailed analysis. Initially, we were unaware that VOC would have been consumed rapidly leading to a significant rise in the radical concentrations especially since these calculations have been performed afterwards (after the experiments were completed). Such calculations are usually not even performed in the studies using OFR. We can finally see how they are beneficial in order to discuss the results obtained. The development of a more detailed model to comprehensively understand the processes involved in the PAM-OFR is needed. In parallel, our objective is to go deeper in the chemical characterization of both, gaseous and particulate phases, using advanced instrumentation such as a CI-ToF-MS (chemical ionization-time of flight - mass spectrometry), combining several ionization modes ($H_3O^+$, $NH_4^+$, $O_2^+$, $I^-$) and aerosol analysis modules [e.g.,

EESI, extractive electrospray ionization (EESI)]. This will help to better understand the chemical processes involved in the formation of SOA inside the PAM-OFR.

Minor comments

Tables 1 and 2: also report precursor concentrations in ppbv to make general comparison a little more straightforward.

Modified accordingly.

Line 145: please provide a sample calculation for each species in the SI.

Furan concentrations were calculated using the syringe pump injection flow rate, temperature (T), analyte molecular weight (MM), density (ρ), and total flow inside the PAM-OFR (Eq. (S1)):

$$[\;]_{ppbv} = \frac{\rho \times pump\ rate\ \times R\ \times T}{MM\ \times P \times Total\ flow} \tag{1}$$

This is now specified in the supplementary material.

Line 158: if the 6-methylchrysene extraction efficiency is higher than 100%, what does this mean for the extraction efficiency of the sampled precursors and calculating precursor concentration? Is the error in this extraction efficiency incorporated into the concentration error bars reported in Tables 1 and 2?

This is a QA/QC step to check the recovery of the applied extraction procedure. As specified in the reference method the acceptance criteria of the recovery of the internal standard is in the range 80 to 120 % (CEN, 2008, 2014). No correction has been applied.

Line 169: add manufacturer details for listed instruments.

Modified accordingly.

Line 192: is the OH exposure listed here the one used to calculate precursor VOC fate (Figure 2) and therefore SOA yield? Or are the KinSim results used?

The $OH_{exp}$ is not an input to the OFR KinSim model. They were obtained after running the model and compared with the experimental OH exposures.

Line 217: the ACSM-derived SOA masses appear to be quite high based on Figure S1. Is there an upper limit of quantification for the Q-ACSM? Did vaporizer performance change over time or was any substantial buildup of material on the vaporizer observed during experiments?

We are not aware of an upper limit of quantification for Q-ACSM. Calibration of the Q-ACSM has been performed at the beginning of the experiments and at the end of the experiments lasting for about 3 months. No drift in the relative ion efficiencies (RIE) obtained for ammonium (=4.58) and sulfate (=1.2) has been observed suggesting that the vaporizer performance did not change over time. Additionally, the Q-ACSM was immediately installed at the ACTRIS SIRTA station to perform ambient air measurements (Haeffelin et al., 2005; Zhang et al., 2019; Petit et al., 2015). All the measurements data have been validated during this period and agreed well with other PM measurements showing again that the high concentrations used during the PAM-OFR experiments did not have any influence on the ACSM measurement and the performance of the vaporizer. Moreover, we have also evaluated the SOA yields based on SMPS measurements (Figure A2). SOA yields showed a good agreement between both approaches (ACSM *vs* SMPS) for most of the precursors. The differences observed might be related to the difference in the size cut between both instruments (Q-ACSM: 110 nm - 3500 ,m; and SMPS: 14.6 - 661.2 nm), and we finally decided to keep only the Q-ACSM data to better consider larger particles that have a significant impact on the PM mass determination and also to avoid some assumptions on the PM density using SMPS data (as we have shown the SOA density varies according to the PM size);

[Figure]

**Figure A2**. Comparison of the PAHs and furans SOA yields obtained from the day- and nighttime oxidation processes (with OH and NO₃ radicals respectively). Results evaluated using Q-ACSM and SMPS measurements and expressed as a function of OH or NO₃ exposures obtained from KinSim simulations. No results are shown for Furan and Flu with NO₃ radicals due to unstable SOA generation or no SOA formed.

Line 221: are similar loss rates expected for larger particle sizes? Many of the experiments produce sizable portions of SOA (both by number and presumably volume) that are in particles with mobility diameter >200 nm (Figure 3).

Yes, similar loss rates were obtained for larger particle sizes up to 300 nm.

Line 270: add x-axes to the upper panels.

Modified accordingly.

Line 372: are heterogeneous oxidation reactions competitive under these OFR conditions?

Heterogeneous reactions may also take place inside the OFR. The sentence has been modified as follows to be clearer (Lines 391-396): "For PAHs, a significant formation of nitro-chromophores species is usually observed upon heterogeneous oxidation processes with $NO_3$ radical (gas/particle) but not for homogenous reactions in the gaseous phase (Cheng et al., 2020; Kwamena and Abbatt, 2008; Lu et al., 2011; Keyte et al., 2013). Once the aerosols formed in the PAM-OFR, heterogenous reactions may occur but reactions involving $NO_3$ interaction with substituent groups through H-atom abstraction, for example for Ace, Acy, and Flu, are not expected to induce an addition of $NO_2$ on the by-products formed (Keyte et al., 2013; Zhou and Wenger, 2013a, b)".

Line 377: why are nitroaromatics not associated with SOA? Keyte et al. (2013) include a discussion of gas-particle partitioning but do not appear to make general conclusions on the fate of nitronaphthalenes or similar compounds.

Overall, several nitroaromatics (such as 2-nitrofluoranthene, 2-nitropyerene, methynitrocathecols, etc…) are associated with SOA. However, nitronaphthalenes and nitronaphthols, produced from naphthalene oxidation (with $NO_3$ in our case), are mainly in the gaseous phase and not associated with the SOA formed. This now specified in the text (lines 396-398): "The reaction of Naph with $NO_3$ alone induces a formation of nitronaphthalenes and nitronaphthols (Keyte et al., 2013) that are mainly in the gaseous phase (Tomaz et al., 2016; Nalin et al., 2016; Reisen et al., 2003) (two-ring compounds) and are not associated with the SOA formed".

**References**

Bhattu, D., Zotter, P., Zhou, J., Stefenelli, G., Klein, F., Bertrand, A., Temime-Roussel, B., Marchand, N., Slowik, J. G., Baltensperger, U., Prévôt, A. S. H., Nussbaumer, T., El Haddad, I., and Dommen, J.: Effect of Stove Technology and Combustion Conditions on Gas and Particulate Emissions from Residential Biomass Combustion, Environ. Sci. Technol., 53, 2209–2219, https://doi.org/10.1021/acs.est.8b05020, 2019.

Bruns, E. A., El Haddad, I., Slowik, J. G., Kilic, D., Klein, F., Baltensperger, U., and Prévôt, A. S. H.: Identification of significant precursor gases of secondary organic aerosols from residential wood combustion, Scientific Reports, 6, 27881, https://doi.org/10.1038/srep27881, 2016.

CEN: European Commitee for Standardization, EN-15549: 2008 - Air Quality - Standard Method for the Measurement of the Concentration of Benzo[a]pyrene in Air. CEN, Brussels (Belgium)., 2008.

CEN: European Commitee for Standardization, TS-16645: 2014- Ambient Air – Method for the Measurement of Benz[a]anthracene, Benzo[b]fluoranthene, Benzo[j]fluoranthene, Benzo[k]fluoranthene, Dibenz[a,h]anthracene, Indeno[1,2,3- cd]pyrene et Benzo[ghi]perylene. CEN, Brussels (Belgium)., 2014.

Cheng, Z., Atwi, K. M., Yu, Z., Avery, A., Fortner, E. C., Williams, L., Majluf, F., Krechmer, J. E., Lambe, A. T., and Saleh, R.: Evolution of the light-absorption properties of combustion brown carbon aerosols following reaction with nitrate radicals, Aerosol Science and Technology, 54, 849–863, https://doi.org/10.1080/02786826.2020.1726867, 2020.

El Mais, A. E. R.: Primary and secondary residential wood combustion emissions: physicochemical and bio-analytical characterization and effect-directed analysis (EDA), PhD Thesis, Université Aix Marseille, 2023.

Haeffelin, M., Barthès, L., Bock, O., Boitel, C., Bony, S., Bouniol, D., Chepfer, H., Chiriaco, M., Cuesta, J., Delanoë, J., Drobinski, P., Dufresne, J.-L., Flamant, C., Grall, M., Hodzic, A., Hourdin, F., Lapouge, F., Lemaître, Y., Mathieu, A., Morille, Y., Naud, C., Noël, V., O'Hirok, W., Pelon, J., Pietras, C., Protat, A., Romand, B., Scialom, G., and Vautard, R.: SIRTA, a ground-based atmospheric observatory for cloud and aerosol research, Annales Geophysicae, 23, 253–275, 2005.

Hatch, L. E., Luo, W., Pankow, J. F., Yokelson, R. J., Stockwell, C. E., and Barsanti, K. C.: Identification and quantification of gaseous organic compounds emitted from biomass burning using two-dimensional gas chromatography–time-of-flight mass spectrometry, Atmospheric Chemistry and Physics, 15, 1865–1899, https://doi.org/10.5194/acp-15-1865-2015, 2015.

Helin, A., Virkkula, A., Backman, J., Pirjola, L., Sippula, O., Aakko-Saksa, P., Väätäinen, S., Mylläri, F., Järvinen, A., Bloss, M., Aurela, M., Jakobi, G., Karjalainen, P., Zimmermann, R., Jokiniemi, J., Saarikoski, S., Tissari, J., Rönkkö, T., Niemi, J. V., and Timonen, H.: Variation of Absorption Ångström Exponent in Aerosols From Different Emission Sources, Journal of Geophysical Research: Atmospheres, 126, e2020JD034094, https://doi.org/10.1029/2020JD034094, 2021.

Keyte, I. J., Harrison, R. M., and Lammel, G.: Chemical reactivity and long-range transport potential of polycyclic aromatic hydrocarbons – a review, Chem. Soc. Rev., 42, 9333, https://doi.org/10.1039/c3cs60147a, 2013.

Kumar, N. K., Corbin, J. C., Bruns, E. A., Massabó, D., Slowik, J. G., Drinovec, L., Močnik, G., Prati, P., Vlachou, A., Baltensperger, U., Gysel, M., El-Haddad, I., and Prévôt, A. S. H.: Production of particulate brown carbon during atmospheric aging of residential wood-burning emissions, Atmospheric Chemistry and Physics, 18, 17843–17861, https://doi.org/10.5194/acp-18-17843-2018, 2018.

Kwamena, N.-O. A. and Abbatt, J. P. D.: Heterogeneous nitration reactions of polycyclic aromatic hydrocarbons and n-hexane soot by exposure to NO3/NO2/N2O5, Atmospheric Environment, 42, 8309–8314, https://doi.org/10.1016/j.atmosenv.2008.07.037, 2008.

Lambe, A. T., Chhabra, P. S., Onasch, T. B., Brune, W. H., Hunter, J. F., Kroll, J. H., Cummings, M. J., Brogan, J. F., Parmar, Y., Worsnop, D. R., Kolb, C. E., and Davidovits, P.: Effect of oxidant concentration, exposure time, and seed particles on secondary organic aerosol chemical composition and yield, Atmos. Chem. Phys., 15, 3063–3075, https://doi.org/10.5194/acp-15-3063-2015, 2015.

Leskinen, J., Hartikainen, A., Väätäinen, S., Ihalainen, M., Virkkula, A., Mesceriakovas, A., Tiitta, P., Miettinen, M., Lamberg, H., Czech, H., Yli-Pirilä, P., Tissari, J., Jakobi, G., Zimmermann, R., and Sippula, O.: Photochemical Aging Induces Changes in the Effective Densities, Morphologies, and Optical Properties of Combustion Aerosol Particles, Environ. Sci. Technol., https://doi.org/10.1021/acs.est.2c04151, 2023.

Li, C., Hu, Y., Chen, J., Ma, Z., Ye, X., Yang, X., Wang, L., Wang, X., and Mellouki, A.: Physiochemical properties of carbonaceous aerosol from agricultural residue burning: Density, volatility, and hygroscopicity, Atmospheric Environment, 140, 94–105, https://doi.org/10.1016/j.atmosenv.2016.05.052, 2016.

Lindberg, J., Wurth, M., Frank, B. P., Tang, S., LaDuke, G., Trojanowski, R., Butcher, T., and Mahajan, D.: Realistic operation of two residential cordwood-fired outdoor hydronic heater appliances—Part 3: Optical properties of black and brown carbon emissions, Journal of the Air & Waste Management Association, 72, 777–790, https://doi.org/10.1080/10962247.2022.2051776, 2022.

Lu, J. W., Flores, J. M., Lavi, A., Abo-Riziq, A., and Rudich, Y.: Changes in the optical properties of benzo[a]pyrene-coated aerosols upon heterogeneous reactions with NO2 and NO3, Phys. Chem. Chem. Phys., 13, 6484–6492, https://doi.org/10.1039/C0CP02114H, 2011.

Nalin, F., Golly, B., Besombes, J.-L., Pelletier, C., Aujay-Plouzeau, R., Verlhac, S., Dermigny, A., Fievet, A., Karoski, N., Dubois, P., Collet, S., Favez, O., and Albinet, A.: Fast oxidation processes from emission to ambient air introduction of aerosol emitted by residential log wood stoves, Atmospheric Environment, 143, 15–26, https://doi.org/10.1016/j.atmosenv.2016.08.002, 2016.

Petit, J.-E., Favez, O., Sciare, J., Crenn, V., Sarda-Estève, R., Bonnaire, N., Močnik, G., Dupont, J.-C., Haeffelin, M., and Leoz-Garziandia, E.: Two years of near real-time chemical composition

of submicron aerosols in the region of Paris using an Aerosol Chemical Speciation Monitor (ACSM) and a multi-wavelength Aethalometer, Atmospheric Chemistry and Physics, 15, 2985–3005, https://doi.org/10.5194/acp-15-2985-2015, 2015.

Reisen, F., Wheeler, S., and Arey, J.: Methyl- and dimethyl-/ethyl-nitronaphthalenes measured in ambient air in Southern California, Atmospheric Environment, 37, 3653–3657, https://doi.org/16/S1352-2310(03)00469-2, 2003.

Schauer, J. J., Kleeman, M. J., Cass, G. R., and Simoneit, B. R. T.: Measurement of Emissions from Air Pollution Sources. 3. C1−C29 Organic Compounds from Fireplace Combustion of Wood, Environ. Sci. Technol., 35, 1716–1728, https://doi.org/10.1021/es001331e, 2001.

Schervish, M. and Donahue, N. M.: Peroxy radical kinetics and new particle formation, Environ. Sci.: Atmos., 1, 79–92, https://doi.org/10.1039/D0EA00017E, 2021.

Srivastava, D., Vu, T. V., Tong, S., Shi, Z., and Harrison, R. M.: Formation of secondary organic aerosols from anthropogenic precursors in laboratory studies, npj Clim Atmos Sci, 5, 1–30, https://doi.org/10.1038/s41612-022-00238-6, 2022a.

Srivastava, D., Vu, T. V., Tong, S., Shi, Z., and Harrison, R. M.: Formation of secondary organic aerosols from anthropogenic precursors in laboratory studies, npj Clim Atmos Sci, 5, 1–30, https://doi.org/10.1038/s41612-022-00238-6, 2022b.

Tomaz, S., Shahpoury, P., Jaffrezo, J.-L., Lammel, G., Perraudin, E., Villenave, E., and Albinet, A.: One-year study of polycyclic aromatic compounds at an urban site in Grenoble (France): Seasonal variations, gas/particle partitioning and cancer risk estimation, Science of The Total Environment, 565, 1071–1083, https://doi.org/10.1016/j.scitotenv.2016.05.137, 2016.

Zhang, Y., Favez, O., Petit, J.-E., Canonaco, F., Truong, F., Bonnaire, N., Crenn, V., Amodeo, T., Prévôt, A. S. H., Sciare, J., Gros, V., and Albinet, A.: Six-year source apportionment of submicron organic aerosols from near-continuous highly time-resolved measurements at SIRTA (Paris area, France), Atmospheric Chemistry and Physics, 19, 14755–14776, https://doi.org/10.5194/acp-19-14755-2019, 2019.

Zhang, Y., Cheng, M., Gao, J., and Li, J.: Review of the influencing factors of secondary organic aerosol formation and aging mechanism based on photochemical smog chamber simulation methods, Journal of Environmental Sciences, 123, 545–559, https://doi.org/10.1016/j.jes.2022.10.033, 2023.

Zhou, S. and Wenger, J. C.: Kinetics and products of the gas-phase reactions of acenaphthene with hydroxyl radicals, nitrate radicals and ozone, Atmospheric Environment, 72, 97–104, https://doi.org/10.1016/j.atmosenv.2013.02.044, 2013a.

Zhou, S. and Wenger, J. C.: Kinetics and products of the gas-phase reactions of acenaphthylene with hydroxyl radicals, nitrate radicals and ozone, Atmospheric Environment, 75, 103–112, https://doi.org/10.1016/j.atmosenv.2013.04.049, 2013b.

Zotter, P., Herich, H., Gysel, M., El-Haddad, I., Zhang, Y., Močnik, G., Hüglin, C., Baltensperger, U., Szidat, S., and Prévôt, A. S. H.: Evaluation of the absorption Ångström exponents for traffic and wood burning in the Aethalometer-based source apportionment using radiocarbon measurements of ambient aerosol, Atmospheric Chemistry and Physics, 17, 4229–4249, https://doi.org/10.5194/acp-17-4229-2017, 2017.

---

## Author Response (AR2)

INERIS
maîtriser le risque |
pour un développement durable |

Verneuil en Halatte, 12th October 2023

Dear Arthur Chan,

Please find below a point-by-point response to your comments (in blue) concerning the manuscript EGUSPHERE-2023-1355 entitled "Insights into secondary organic aerosol formation from the day- and nighttime oxidation of PAHs and furans in an oxidation flow reactor". We have addressed your comments and revised the manuscript accordingly. We think that this new version can now fully meet the standards of the Atmospheric Chemistry and Physics journal.

With very best wishes,

Dr. (HDR) Alexandre ALBINET

INERIS

Parc Technologique ALATA – BP 2

Verneuil-en-Halatte, 60550 (France)

Email: alexandre.albinet@gmail.com; alexandre.albinet@ineris.fr

Please refer to the author guidelines: https://www.atmospheric-chemistry-and-physics.net/policies/guidelines_for_authors.html

Specifically, the abstract is too long, and the conclusions should include a more thorough discussion of context, limitations and implications.

The abstract has been shortened to fit with ACP's guidelines. The conclusions have been modified and now include a better introduction of the context as well as a discussion on the limitations and implications of the work.

Also, obtaining results for other studies (biological assessments) is not necessarily a justification for the high loadings in this study. This manuscript is a standalone work and the conditions should be relevant for the research questions in this particular study. The loadings are high even for a laboratory SOA study. The other justification about being relevant to near-field conditions is not that strong either. If so, more evidence should be presented about how relevant that would be (e.g. how long would it take for the plume to dilute to lower PM levels). In general, I do not believe the authors have quite addressed a major concern raised by almost all the reviewers, and a better justification is needed.

We totally agree that the high SOA loadings used in this study are not relevant to ambient air conditions. However, they are relevant to the OFR ageing studies performed on biomass burning emissions as well as solid fuel combustion emissions and to the study of the subsequent formation of secondary particles (comparable either in terms of VOC concentrations or resulting PM/OA concentrations in the OFR) (e.g. Bruns et al., 2015; Budisulistiorini et al., 2021; Iaukea-Lum et al., 2022; Ortega et al., 2013; Reece et al., 2017; Zhang et al., 2021). The list here is not exhaustive and most of the studies performed on such combustion sources have been probably done using high loadings (the dilution factors applied are not high enough to reach ambient air conditions). Such studies are performed to be further extrapolated to ambient air and are relevant to address the research questions related to atmospheric chemistry or atmospheric processes. In addition, such high concentration conditions used in this type of studies are also related to potential certification purposes of the particulate emissions of residential heating appliances or combustion engines (Cao et al., 2020).

Moreover, several previous OFR studies focusing on the SOA from pure anthropogenic precursors, and addressing similar research questions, have been also performed in similar high loading conditions (e.g. Lau et al., 2021; Liu et al., 2015).

Finally, as specified before, toxicological studies on SOA based on the use of OFR also used usually high precursor concentrations to get sufficient amount of PM material (e.g., Wang et al., 2018; Khan et al., 2021; Offer et al., 2022).

This is now specified in the text and the text has been modified as follows:

[revised manuscript text omitted]